# Empirical and model-based evidence for a negligible role of cattle in peste des petits ruminants virus transmission and eradication
Catherine M. Herzog [1] ✉, Fasil Aklilu [2,11], Demeke Sibhatu[2], Dereje Shegu [2], Redeat Belaineh[2], Abde Aliy Mohammed[2], Menbere Kidane[2], Claudia Schulz [3], Brian J. Willett [4], Sarah Cleaveland [5], Dalan Bailey [6], Andrew R. Peters[7], Isabella M. Cattadori [1], Peter J. Hudson [1], Hagos Asgedom[2], Joram Buza[8], Mesfin Sahle Forza[2,10], Tesfaye Rufael Chibssa[2], Solomon Gebre[2], Nick Juleff [9], Ottar N. Bjørnstad [1], Michael D. Baron[6] & Vivek Kapur [1] ✉

Peste des petits ruminants virus (PPRV) is a multi-host pathogen with sheep and goats as main hosts. To investigate the role of cattle in the epidemiology of PPR, we simulated conditions similar to East African zero-grazing husbandry practices in a series of trials with local Zebu cattle (*Bos taurus indicus*) co-housed with goats (*Capra aegagrus hircus*). Furthermore, we developed a mathematical model to assess the impact of PPRV-transmission from cattle to goats. Of the 32 cattle intranasally infected with the locally endemic lineage IV strain PPRV/Ethiopia/Habru/2014 none transmitted PPRV to 32 co-housed goats. However, these cattle or cattle co-housed with PPRV-infected goats seroconverted. The results confirm previous studies that cattle currently play a negligible role in PPRV-transmission and small ruminant vaccination is sufficient for eradication. However, the possible emergence of PPRV strains more virulent for cattle may impact eradication. Therefore, continued monitoring of PPRV circulation and evolution is recommended.

A challenge in the control and eradication of many infectious diseases is the ability of several pathogens to infect multiple host species, some of which can act as reservoirs of infection[1]. Pathogen control programs need to identify the role of each host species in transmission dynamics and to determine which host species permit pathogen persistence within the host community[2,3]. Tackling this challenge requires leveraging multiple tools simultaneously: experiments that shed light on host species susceptibility and transmission competency, models that capture our understanding of the dynamics of transmission and explore outcomes of potential intervention scenarios, and the collection of epidemiological and ecological field data that inform and improve the models. Such combined approaches have been used effectively for rabies to disentangle the role of multiple hosts in viral

persistence[4], to identify specific mechanisms that alter transmission at varying spatial scales[5], and to inform progress toward control and elimination[6,7]. If eradication is the goal but longitudinal field studies are challenging, an interdisciplinary approach that combines both empirical and modeling scenarios may be the only feasible approach for progress. This is the case for the economically important, multi-host disease caused by the livestock pathogen peste des petits ruminants virus.

Peste des petits ruminants (PPR) virus (PPRV; family: *Paramyxoviridae*; genus: *Morbillivirus*; species: *Small ruminants morbillivirus*) is a priority pathogen that is targeted for global eradication by 2030 by the Food and Agricultural Organization (FAO) and the World Organization for Animal Health (WOAH)[8,9]. PPRV threatens nearly 80% of the global

[1]Center for Infectious Disease Dynamics, Pennsylvania State University, University Park, PA, USA. [2]Animal Health Institute (AHI), Sebeta, Ethiopia. [3]Institute of Virology, Department of Biological Sciences and Pathobiology, University of Veterinary Medicine Vienna, Vienna, Austria. [4]MRC-University of Glasgow Centre for Virus Research, College of Medical, Veterinary and Life Sciences, University of Glasgow, Glasgow, UK. [5]School of Biodiversity, One Health and Veterinary Medicine, College of Medical, Veterinary and Life Sciences, University of Glasgow, Glasgow, UK. [6]The Pirbright Institute, Pirbright, UK. [7]Supporting Evidence Based Interventions (SEBI), University of Edinburgh, Edinburgh, UK. [8]Nelson Mandela African Institute of Science and Technology, Arusha, Tanzania. [9]Bill & Melinda Gates Foundation, Seattle, WA, USA. [10]Deceased: Mesfin Sahle Forza. [11]These authors contributed equally: Catherine M. Herzog, Fasil Aklilu. ✉e-mail: cqh5447@psu.edu; vkapur@psu.edu

small ruminant population and the livelihoods of over 330 million farmers[10]. The virus can cause high morbidity and moderate to high mortality rates in the main hosts, domesticated goats and sheep. PPRV is typical among morbilliviruses in that it infects a range of livestock and wildlife host species, with varying levels of host pathogenicity[11], and results in robust immune memory in recovered hosts. Information on host range has come from observed outbreaks, serological data, and a small number of experiments[12–18] in white-tailed deer, pigs, and camels which directly address the ability of each species to transmit PPRV. Importantly, there is concern that the proposed eradication efforts, which based on mass vaccination of small ruminants, may be thwarted if PPRV has the potential to persist within currently unidentified livestock or wildlife reservoirs[5,9]. The turnover of sheep and goat populations is high and maintaining high immunity at a population scale may be challenging. Mass vaccination of sheep and goats alone could be insufficient to eradicate PPR and the potential for spillback events into susceptible host populations would remain. Therefore, there is an urgent need to better understand the role of a larger number of livestock and wildlife species in PPR epidemiology and to identify what additional control measures may be needed to account for these atypical host species.

Cattle are an important focus for investigation as they are abundant, live in close proximity to sheep and goats, and serological studies indicate that they are frequently exposed to PPRV infection. Previous cross-sectional studies found widespread PPRV seroconversion in cattle in Africa and Asia[19–33]. Experimentally infected cattle seroconvert[34,35] and shed PPRV RNA detectable from 1–10 days post infection (dpi)[35], but not infectious virus[34,35]. However, of these studies, one did not use local cattle breeds from the same region where the PPRV strain was isolated[35], which limits the conclusions that can be drawn around transmission and susceptibility of locally-circulating PPR viruses and hosts. The second study did not determine if infectious virus was present in cattle and used a host sample size that would only be able to detect infection if a high rate of cattle-to-goat transmission was present (i.e., $n = 12$, with 24 possible cattle-goat contacts)[34]. Hence, it remains unclear if PPRV-infected cattle are capable of shedding PPRV at a level sufficient for transmission to sheep and goats or other cattle. Whether cattle can transmit PPRV to small ruminants in experimental conditions and understanding the scenarios of inter-species transmission would enable PPRV to persist in the field are key questions that need to be answered to resolve the role of this atypical host.

The objectives of the current study were twofold: first, to establish through experiments whether local cattle infected with locally circulating PPRV strains shed and transmit PPRV to seronegative (naïve) small ruminants and second, to use a mathematical modeling approach to quantify the extent to which varying scenarios of potential cattle transmission rates could impact PPR eradication efforts. This was achieved through a series of PPRV transmission trials in Ethiopia using a local PPRV isolate and local breeds of cattle, sheep, and goats. The trials were designed to emulate mixed species zero-grazing husbandry practices commonly observed in East Africa, where PPR is endemic. We hypothesize that cattle are capable of transmitting PPRV to small ruminants and thus contribute to PPRV persistence in the field. However, since cattle-to-small ruminant transmission was expected to be rare, this study was designed to be able to detect a low rate of cattle-to-small ruminant transmission and define an expected upper limit on the probability of such transmission. We developed a cross-species susceptible-infected-recovered (SIR) model with small ruminant vaccination to estimate the impact of cattle transmission on the effective community reproductive number, $R_e$, and to determine the proportion of vaccinated small ruminants needed to control community transmission under varying cattle-to-goat transmission scenarios. Together, the experimental trials and modeling framework suggest that cattle are currently not playing a significant role in the transmission of PPRV to sheep and goats and that there is no need to modify the PPR eradication campaign to include control measures for cattle.

## Materials and methods

### Virus
The PPRV lineage IV (LIV) isolate was collected in 2014 from pooled samples of a single female goat in the Habru district in the Oromia Zone of the Amhara region[36]. This isolate caused 21.9% morbidity and 8.4% mortality with a case fatality rate of 38.4% in a population of 511 (121 sheep, 390 goats)[36]. After being passaged twice on CHS-20 cells, which are (Flp-In-CV1) CV1 African Green Monkey kidney cells stably expressing the goat SLAM receptor[37], the isolate PPRV/Ethiopia/Habru/2014 was stored at the Animal Health Institute (AHI, formerly National Animal Health Diagnostic Investigation Center - NAHDIC) at −80 °C[36]. For the trials, the virus was further grown on Vero-dogSLAM (VDS) cells[38] and harvested starting at the third passage (Table S1). The full genome sequence was determined at VDS passage 3 and 4 (Genbank accession ON110960) and no difference was found between these two preparations. Tissue culture infectious dose ($TCID_{50}$) was calculated using the Spearman–Kärber Method[39].

### Animals and study design
Local breeds of sheep (*Ovis aries*), goats (*Capra aegagrus hircus*), and Zebu calves (*Bos taurus indicus*) were purchased from markets within a 200 km radius of AHI (Sebeta, Ethiopia). There was no reported history of PPRV vaccination in any animal purchased. Animals were of both sexes and ranged in age from 6 months to 1.5 years. Once seronegative status was confirmed (see Serological Analysis), animals were moved into the six-barn experimental facility for acclimatization, followed by randomization of experimental animals and controls to specific experimental barns (also randomized). More details on animal recruitment, sampling, barn conditions, biosecurity, and positive and negative control animals are provided in the supplement (Text S1–S3). The cattle-to-goat transmission trial was designed with the necessary sample size to have an 80% chance of detecting a cattle-to-goat transmission event, if it occurred, for an event probability as low as 0.05 (Text S4, Fig. S1). We have complied with all relevant ethical regulations for animal use (see Inclusion & Ethics Statement for details).

A series of 5 trials were conducted (Fig. S2) in which sheep or goats and cattle were infected intranasally with 1 ml of viral isolate in Dulbecco's Modified Eagle's Medium (DMEM), 0.5 ml per nostril. Cattle were given 2 ml total virus, 1 ml per nostril. Specifically, passage 3 or 4 was used, and $TCID_{50}$ ranged between $10^{5.3}$ and $10^{5.6}$. Virus titer and passage for each trial are in Table S1. For transmission trials, sentinel (seronegative) animals were added to the barn 1 day post-infection (dpi). Within the barn, animals were allowed to move freely and shared feeding and watering troughs (Fig. S3). Each day animals were monitored for clinical signs using an established clinical scoring scheme which included rectal temperature[35,40]. Each of the 6 component scores ranged from 0–4 and were summed to the final score. Samples (whole blood, serum, and ocular, nasal, and rectal swabs) were collected from every animal on 0, 4, 7, 10, 14, 17, 21, 28 dpi. For trials 3–5, sampling days were extended to include 32 and 35 dpi to better understand pathogen shedding in animal feces. Total animals used in each barn and each trial are listed in Fig. S2, and in the conceptual figured inset in the upper left of each results figure.

### Clinical validation of isolate
Trials 1 and 2 established that a single 1 ml intranasal dose of the local isolate PPRV/Ethiopia/Habru/2014 produced typical clinical signs in local breeds of sheep and goats and transmitted readily from sheep to sheep and goat to goat (Figs. S4–S9, Text S1). Sheep were not used after Trial 2. A supplemental PPRV challenge trial confirmed that seroconversion protected goats from clinical disease (Fig. S9, Text S1). Results from trials 1–2 are reported in the supplement and we focus here only on the investigation of cattle-to-goat transmission (trials 3–5). For each trial, new positive (inoculated) and negative (PPRV seronegative) control animals were run at the same time, in the same facility, in a randomly assigned barn (Figs. S10–S12).

## Serological analyses

Goat, sheep, and cattle sera were tested in duplicate for antibodies against the PPRV N protein using competitive ELISA[41] (sensitivity: 94.5%, specificity: 99.4%) according to the manufacturer's instructions (Text S2).

## Molecular analyses

PPRV RNA from swabs and whole blood taken on each sampling day was extracted and PPRV RNA assayed using quantitative reverse transcription real-time PCR (qRT-PCR) based on Batten et al.[42] with custom Taqman QSY Probe and Express One Step kit (Text S2). Samples were tested in duplicate and those with a cycle threshold (Ct) value of 35 or less were considered positive.

Swabs were also tested by commercial antigen sandwich ELISA kit (Text S2) and included nasal swabs from 4 dpi + and rectal swabs from 14 dpi + (trials 1–3) or 4 dpi+ (trials 4–5). Ocular swabs were not tested. This test detected PPRV nucleoprotein using anti-PPRV N protein antibodies. For trials 2-5, the qRT-PCR data were compared with the antigen ELISA results (Figs. S8–S13). The commercial kit had not been validated in cattle.

Swab samples were not taken from negative control animals after trial 1 and they were monitored solely for clinical signs at a daily interval and seroconversion on sampling days. Whole blood was not tested for PPRV RNA after trial 2.

## Virus isolation

Ocular, nasal, or rectal swab samples that tested positive by qRT-PCR or antigen sandwich ELISA were tested for infectious virus by inoculating into Vero-dogSLAM (VDS) cells[38] and recorded as PPRV-positive if typical CPE was observed within 5 days. Details in Text S2.

## Statistics and reproducibility

For each trial, plots of rectal temperature, clinical score[35,40], serology, viral RNA and or viral protein/antigen from nasal swabs over time (dpi) were created using the ggplot package in R statistical software 4.3.3. Local regression (LOESS) smoothed curves and accompanying 95% confidence bands (from t-based approximation) were added in bold to the plots using the stat_smooth function. These smoothed lines represent all animals in the specific group (control, inoculated, sentinel). The dpi at which the peak value for each metric in each animal group was achieved was extracted and plotted as a vertical line corresponding to the color of the group and an inter-peak interval was calculated by taking the difference in dpi of the peak values (sentinel peak dpi – inoculated peak dpi).

Using the trial results in a beta-binomial Bayesian model from the bayesrules package in R[43,44], posterior distributions were estimated for the following transmission probabilities: cattle-to-small ruminants ($\beta_{CS}$), small ruminants-to-cattle ($\beta_{SC}$), and small ruminant-to-small ruminant ($\beta_{SS}$). These reported probabilities represent a net transmission from one species to another in the trials. The individual animal transmission probability is calculated by dividing the posterior mean by the number of inoculated animals. Transmission probabilities were converted into rates using the following relationship:

$$p(t) = 1 - e^{-rt}$$

where p(t) is the transmission probability, r is the transmission rate, and t represents the infectious period (10 days[35,45] used in main text, results with 8 and 14 available in supplement). Credible intervals-(CrI) were calculated for the posterior mean and transmission rates using the appropriate quantile functions in R (qbeta or rbeta for sampling and then quantile). Calculated transmission rates were used as transmission rate parameters between animal species ($\beta_{CS}$, $\beta_{SC}$, $\beta_{SS}$) in the mathematical model. Code is available in Text S5 and on GitHub (see Data Accessibility) and visualization of the priors and posterior distributions in Fig. S14.

## Mathematical modeling

To explore the impact of potential cattle transmission to small ruminants under the current eradication plan in which only small ruminants are vaccinated, a Susceptible-Infected-Recovered (SIR) dynamical model was developed and applied to multiple species.

$$\frac{d\vec{S}}{dt} = \underbrace{\mu\vec{N}(1 - \vec{p})}_{\text{recruitment}} - \underbrace{\phi\vec{S}}_{\text{infection}} - \underbrace{\mu\vec{S}}_{\text{death}}$$

Force of infection : $\phi = \beta \times (\vec{I}/\vec{N})$

$$\frac{d\vec{I}}{dt} = \underbrace{\phi\vec{S}}_{\text{infection}} - \underbrace{\gamma\vec{I}}_{\text{recovery}} - \underbrace{\mu\vec{I}}_{\text{death}}$$

$$\frac{d\vec{R}}{dt} = \underbrace{\gamma\vec{I}}_{\text{recovery}} - \underbrace{\mu\vec{R}}_{\text{death}}$$

In the above system of equations, which represents species-specific populations (N), vaccination of the susceptible sheep or goats is included (1 - p), as well as birth ($\mu$N), death ($\mu$S, $\mu$I, $\mu$R), and recovery ($\gamma$I). Cattle were not vaccinated ($p_{cattle} = 0$). The parameter p represents small ruminant vaccination coverage and effectiveness combined. This model assumes homogeneous mixing of hosts.

Using the next-generation framework[46], the community reproductive number ($R_0$) – the average number of secondary infections produced by one infected animal—was calculated over a range of small ruminant vaccination (0–100%) and cattle-to-small ruminant transmission rates ($\beta_{CS}$ = 0-slightly above the rate of goat-to-goat transmission). Parameter values are found in Table S2. Mortality rates were drawn from the literature[47] and the inverse of the rate yielded reasonable life expectancy for small ruminants (2.14 years) and cattle (10 years). Recovery from PPRV infection takes approximately 14 days; the recovery rates ($\gamma$) for small ruminants and cattle were assumed the same (1/14). Transmission rates between each species ($\beta_{CS}$, $\beta_{SC}$, $\beta_{SS}$) were calculated from the posterior distributions of the transmission probabilities from the trials. As control was applied and small ruminant population immunity changed, the values of the effective community reproductive number $R_E$ were estimated. Four plausible transmission patterns (two symmetric, two asymmetric) were explored as the cattle transmission rate ($\beta_{CS}$; where C = cattle and S = small ruminants) was varied, but other values in the 2 × 2 transmission parameter β matrix remained fixed. Specifically, scenario 1 explored a symmetric transmission rate for all species ($\beta_{SS} = \beta_{SC} = \beta_{CC}$; $\beta_{CS}$ varies 0–0.26), scenario 2 explored a symmetric transmission rate within each species ($\beta_{SS} = \beta_{SC}$; $\beta_{CC}$; $\beta_{CS}$ varies 0–0.26), scenario 3 explored an asymmetric transmission rate between species ($\beta_{SS}$; $\beta_{SC} = \beta_{CC}$; $\beta_{CS}$ varies 0–0.26), and scenario 4 explored an asymmetric transmission rate within and between species ($\beta_{SS}$; $\beta_{SC}$; $\beta_{CC} = 0$; $\beta_{CS}$ varies 0–0.26). R software code for the model is provided in Text S6 and on GitHub (see Data Accessibility).

## Reporting summary

Further information on research design is available in the Nature Portfolio Reporting Summary linked to this article.

## Results and discussion

### Cattle can be infected naturally by contact with PPRV infected goats and can show mild, transient clinical signs

The PPRV/Ethiopia/Habru/2014 isolate infected local Zebu cattle under natural conditions. Two PPRV-naïve cattle co-housed with 4 inoculated goats seroconverted, though these cattle seroconverted later than 2 PPRV-

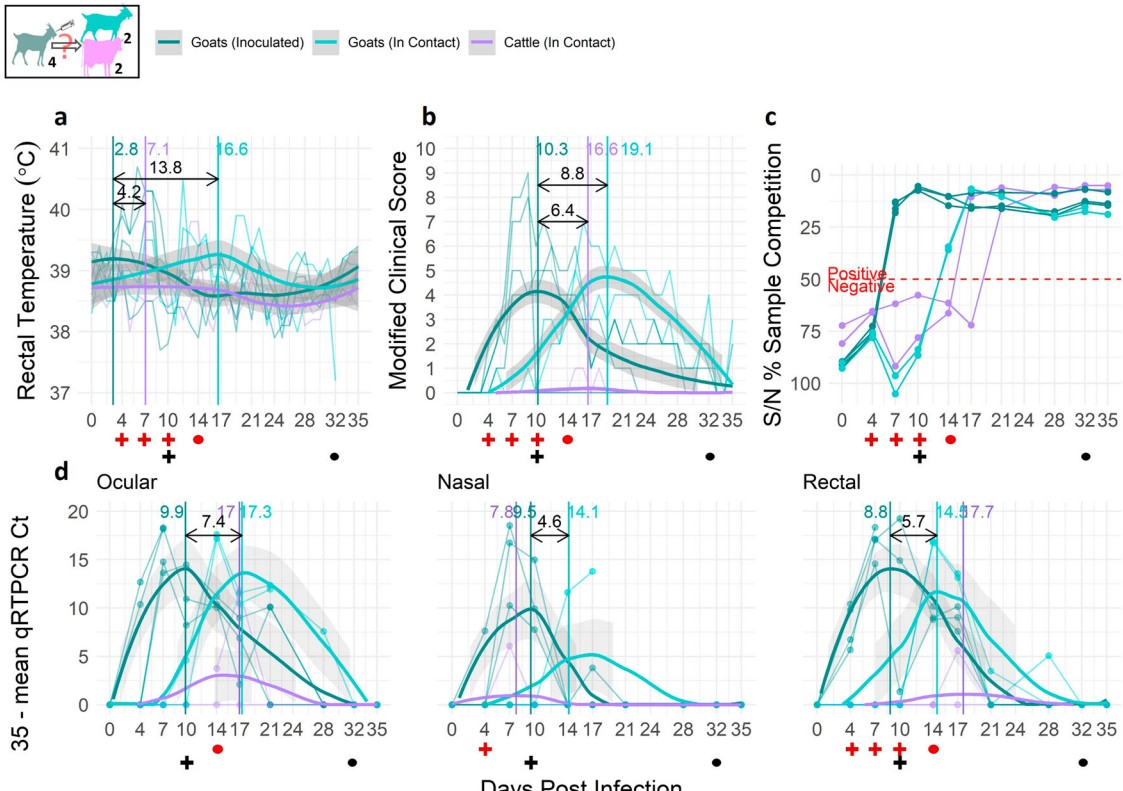

**Fig. 1 | Cattle are naturally infected and seroconvert after contact with co-housed PPRV infected goats. a** daily rectal temperature, **b** daily clinical score, **c** serology (competitive ELISA), and **d** viral RNA from ocular, nasal, and rectal swabs (qRT-PCR). Thin lines represent individual animals and bold lines represent smooth local regression (LOESS) curves of all animals in the category (control, inoculated, sentinel). Gray shading indicates 95% confidence bands (t-based approximation). Vertical lines indicate day post infection (dpi) of peak value and inter-peak interval

(difference in dpi of peak value for each animal group) is indicated in black. Sampling days on which PPRV could be isolated (cross if isolated from inoculated animal, circle for sentinel animal) are indicated with red and dpi with deaths are indicated with black (cross for inoculated animal deaths, circle for sentinel deaths, x for euthanized) along the x-axis. No PPRV was isolated from cattle samples. Two deaths occurred, one inoculated and one sentinel goat. Image Icon Credits: Get-Drawing.com (goat[73], cattle[74]), Clker.com (needle[75]).

naïve goats co-housed in the same barn (17–21 dpi vs 14 dpi, respectively; Fig. 1). In this trial, days were counted starting with goat inoculation, so cattle were exposed to the excreted virus only from 4–5 dpi. Calf 1 had no clinical signs. Calf 2 had only mild, transient clinical signs between 13–18 dpi, including slightly elevated rectal temperature (≥ 39.5 °C but ≤ 40 °C) and displayed mildly inactive, tired behavior. PPRV RNA was detected by qRT-PCR in both replicates of nasal (7 dpi) and rectal (17 dpi) swabs, and one replicate of ocular (17 dpi) swabs from Calf 2 (Fig. 1). These positive qRT-PCR samples (all Ct > 27) were negative when virus isolation in cell culture was attempted. When testing for viral antigens by AgELISA, Calf 2 had positive nasal swabs on 7 and 10 dpi and positive rectal swabs on 21, 28, and 32 dpi. Calf 1 had positive nasal swabs on 7, 10, 17, and 21 dpi but did not have positive rectal swabs. Antigen results from these sentinel calves are not visualized.

This experiment demonstrated that calves can be infected naturally with PPRV by contact with infected goats and show mild, transient clinical signs. Based on this, we suggest that cattle in the field are most likely exposed to PPRV as calves (<1 year) through direct contact when co-housed with sheep and goats or when a PPR epidemic spreads through a co-housed or communally grazed herd after new animals are introduced. Previously published studies reported low seroconversion rates among cattle in the field across several African and Asian countries (1.8–18%, n = 24–1158)[19–33] which may be due to few encounters with PPRV-infected sheep or goats by management practice or chance, pre-existing cross-immunity from infections with other circulating morbilliviruses[48], or infection with PPRV strains that are either able to evade[49] or are unable to overcome the cattle innate immune response—both of which might reduce seroconversion.

In contrast, two field studies from Pakistan and Sudan report high PPRV seroprevalence among cattle (41.9–42%, n = 43–1000)[20,23]. Animals sampled either experienced[20] or were hypothesized to have experienced[23], shared grazing and watering. Additional explanations for high seroprevalence could include: cattle have longer lifespans, relative to sheep and goats, and so have a greater chance of PPRV exposure and seroconversion; an ongoing source of PPRV exposure is present such as frequent contact with newly introduced PPRV-infected animals, extended shedding of PPRV from infected sheep and goats, or extended PPRV survival in some environments. More data is needed to support any of these explanations. Future work could vary the number of co-housed infectious goats as a proxy for varying PPRV dose to better understand the minimum PPRV dose that consistently would produce clinical signs in cattle.

## Cattle experimentally infected with PPRV do not develop clinical disease and do not transmit PPRV to goats

Cattle-to-goat transmission trials were conducted to determine if inoculated cattle could transmit PPRV to co-housed goats. Intranasally inoculated cattle seroconverted between 7 to 14 dpi (21.9% by dpi 7 and 93.7% by dpi 10, Fig. 2), developed no pyrexia and no, or only mild, transient clinical signs. None of the inoculated cattle shed PPRV RNA as detected by qRT-PCR. Nasal and rectal swabs from the inoculated cattle were positive for PPRV antigens as detected by AgELISA (Fig. S13), but antigen presence was transient and at lower levels than infected goats (Fig. S8). Low levels of antigen were detected in cattle nasal swabs up to 21 dpi (11 of 12 detected up to 10 dpi) and rectal swabs up to 32 dpi (Fig. S13), which contrasted with the clear peak of antigen detection in infected goats, which resolved by 17 dpi (Fig. S8).

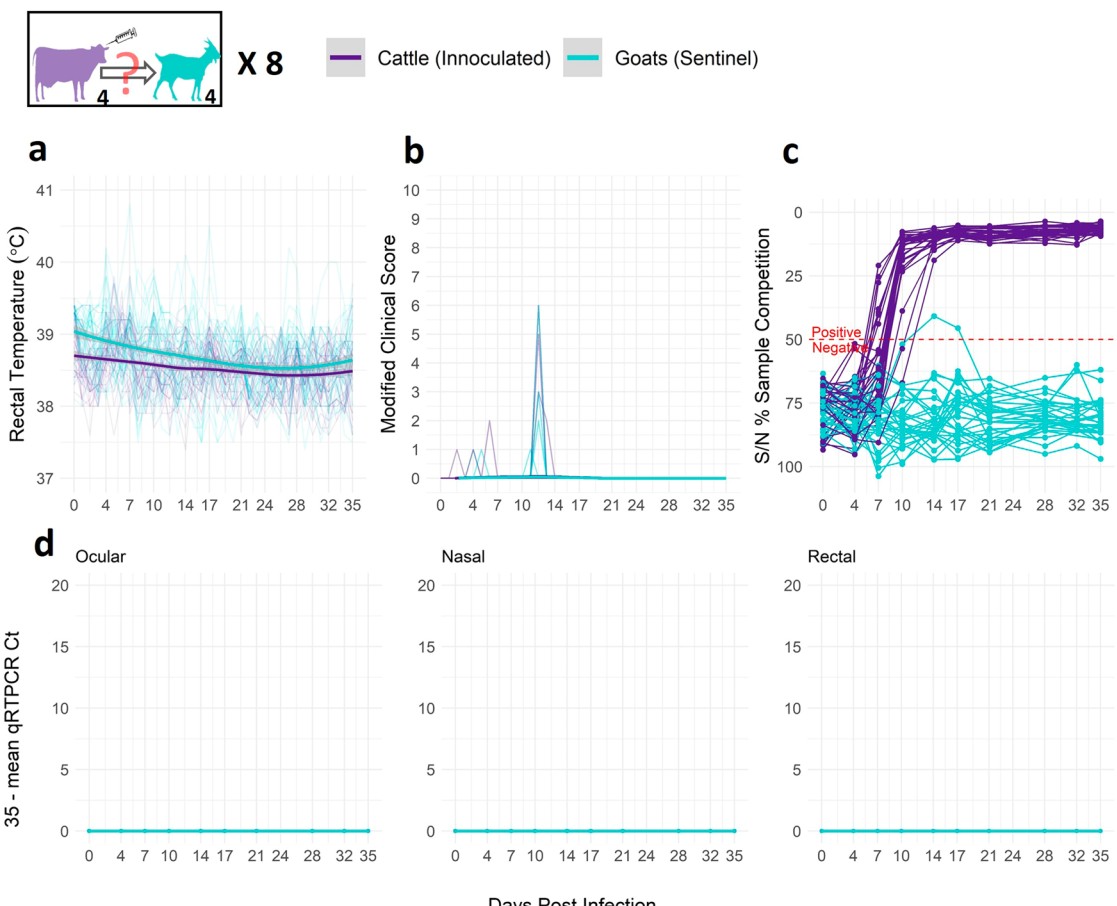

**Fig. 2 | Inoculated cattle do not transmit PPRV to co-housed sentinel goats.**
**a** daily rectal temperature, **b** daily clinical score, **c** serology (competitive ELISA), and
**d** viral RNA from ocular, nasal, and rectal swabs (qRT-PCR). Each trial used 4th
passage isolate $TCID_{50}$ $10^{5.5-5.6}$. Thin lines represent individual animals and bold
lines represent smooth local regression (LOESS) curves of all animals in the category
(control, inoculated, sentinel). Gray shading indicates 95% confidence bands (t-
based approximation). No deaths occurred (with the exception of the death of one
goat due to injury unrelated to infection) and no isolates were recovered from cattle.
Image Icon Credits:GetDrawing.com (goat[73], cattle[74]), Clker.com (needle[75]).

The PPRV/Ethiopia/Habru/2014 isolate transmitted efficiently among
sheep and among goats (Fig. S5) and from inoculated goats to cattle (Fig. 1):
all 12 sentinel experimental animals became infected and all 4 sentinel
positive control goats. The small ruminant-to-small ruminant transmission
probability ($9.29 \times 10^{-1}$, 95% CrI: $7.53 \times 10^{-1}$, $9.98 \times 10^{-1}$) was 1.24 times
the small ruminant-to-cattle transmission probability ($7.50 \times 10^{-1}$, 95% CrI:
$2.92 \times 10^{-1}$, $9.92 \times 10^{-1}$), and the calculated small ruminant-to-small
ruminant transmission rate was nearly double the small ruminant-to-
cattle transmission rate ($\beta_{SS} = 2.64 \times 10^{-1}$ (95% CrI: $1.44 \times 10^{-1}$,
$6.33 \times 10^{-1}$); $\beta_{SC} = 1.39 \times 10^{-1}$ (95% CrI: $3.45 \times 10^{-2}$, $4.87 \times 10^{-1}$)). In con-
trast, no evidence was found for PPRV transmission from inoculated cattle
to co-housed goats (Fig. 2). Co-housed goats developed no lasting PPRV
antibodies as detected by cELISA, shed no virus detectable by qRT-PCR, and
showed minimal clinical signs not consistent with a typical course of disease
(e.g., transient days of more inactive behavior up to score 2, and/or changes
of score 1 in feces, discharges, or respiration[40]). Each barn contained 4
inoculated cattle and 4 sentinel co-housed goats, giving 16 potential trans-
mission contacts per barn; across all cattle-to-goat trials, this was replicated 8
times giving a total of 128 potential transmission contacts. With no cattle-to-
goat transmission detected among the 32 inoculated cattle and 32 co-housed
goats, this study provides strong evidence that the probability of cattle-to-
goat transmission is rare, including when animals are co-housed for an
extended period (Fig. S1). This is supported by the nearly 32 fold difference
in the estimated posterior mean of the small ruminant-to-small ruminant
transmission probability ($9.29 \times 10^{-1}$, 95% CrI: $7.53 \times 10^{-1}$, $9.98 \times 10^{-1}$)
compared to the cattle-to-goat transmission probability ($2.94 \times 10^{-2}$; 95%

CrI: $7.67 \times 10^{-4}$, $1.06 \times 10^{-1}$), which translates to an 88 fold difference in
transmission rates ($\beta_{SS} = 2.64 \times 10^{-1}$ (95% CrI: $1.44 \times 10^{-1}$, $6.33 \times 10^{-1}$),
$\beta_{CS} = 2.99 \times 10^{-3}$ (95% CrI: $8.60 \times 10^{-5}$, $1.13 \times 10^{-2}$) (Text S5, Fig. S14).
Serological findings in this study can be compared to two smaller
trials[34,35]. Specifically, this study found cattle seroconverted between 7–14
dpi, whereas Schulz et al.[35] and Couacy-Hyman et al.[34] documented ser-
oconversion at 10 dpi and 9 or 15 dpi (depending on the isolate), respec-
tively. Unlike Schulz et al.[35], this study found no PPRV RNA by qRT-PCR
from any swab of an inoculated cattle. However, PPRV RNA was detected
by qRT-PCR in one naturally infected sentinel calf, but PPRV was not
isolated from these samples. This finding suggests it may be important to
continue to monitor PPRV in cattle and to further investigate the
mechanisms that prevent PPRV-induced pathogenesis and viral excretion
in cattle, such as differences in viral localization[35], receptor usage[50], inter-
cellular transmission pathways, or replication dynamics in lymph nodes and
immune compartments vs the rest of the cattle host[51] as well as if these
mechanisms vary with PPRV strain or cattle breed. Lastly, this study
detected PPRV antigens in nasal and rectal swabs by AgELISA among cattle
in contact with inoculated goats; however it is highly likely that the early
positive nasal swabs (7 and 10 dpi) represents goat-excreted virus that the
calf inhaled and that positive rectal swabs represents dead tissue shedding as
opposed to live virus. Inoculated cattle positive nasal swabs (4, 7, 10, 21 dpi;
Fig. S13) were likely due to remaining PPRV from nasal inoculation, but
could potentially also be from local replication in nasal epithelial cells as has
been observed with measles and canine distemper viruses[52]. We note as well
that the AgELISA commercial assay used in this study has been validated for

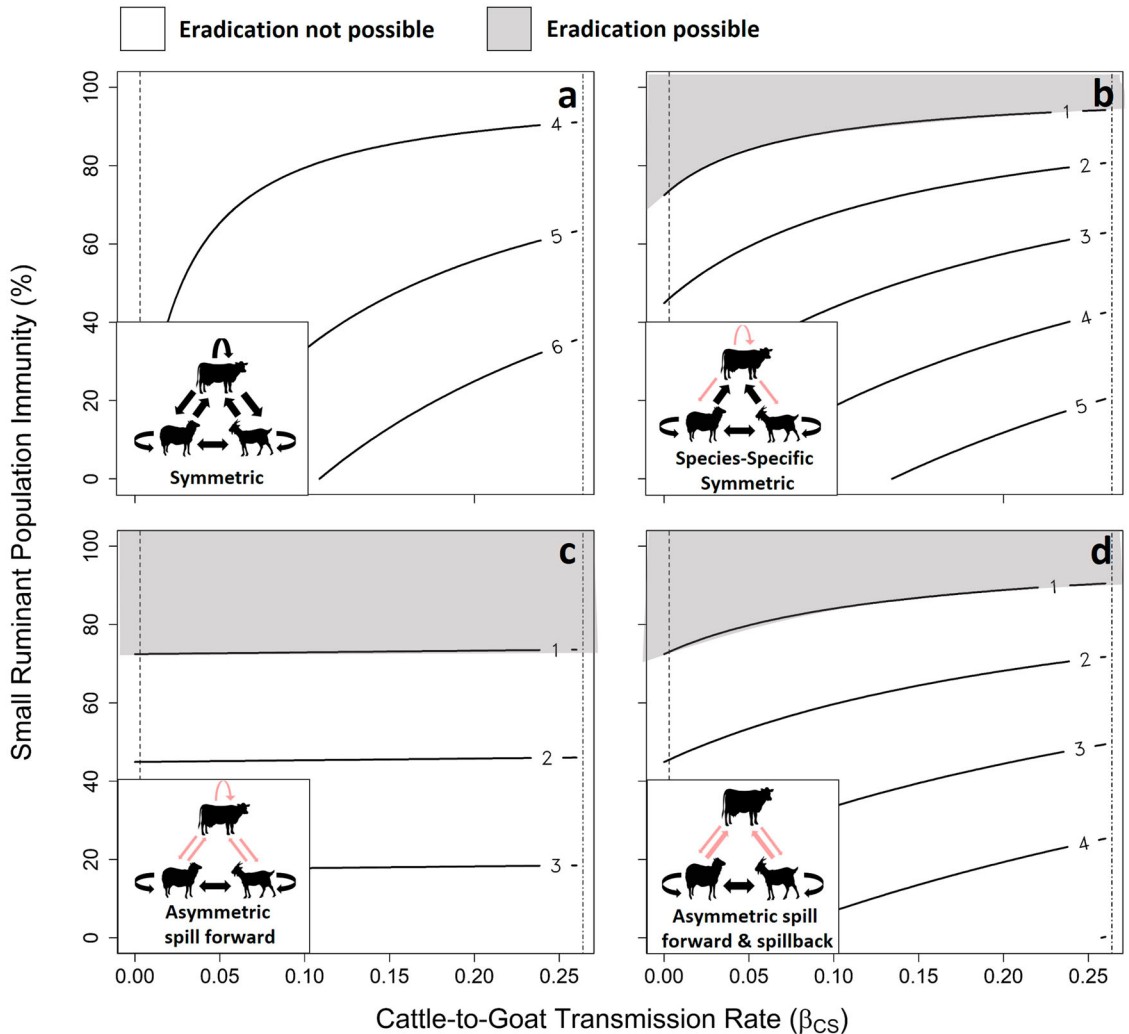

**Fig. 3 | Increasing cattle-to-small ruminant transmission increases the amount of small ruminant vaccination needed in some but not all transmission scenarios.** Eradication is possible in scenarios B, C, and D when $R_E < 1$ (i.e., area above the contour labeled "1" in B, C & D and shaded in gray), but is not possible in scenario A. Community $R_E$ values are indicated in the numeric values on labeled contours. For an infectious period of 10 days, the estimated transmission rates for cattle-to-small ruminant and small ruminant-to-small ruminant are shown as dashed (farther left) and dot dashed (farther right) vertical lines, respectively. In scenarios B and D, increasing levels of small ruminant vaccination is indicated, as the rate of cattle-to-small ruminant transmission increases from the observed cattle-to-goat transmission rate to slightly above the goat-to-goat transmission rate observed in the trials. For each transmission scenario, cattle (C) transmission rate to small ruminants (S) varies from 0 to 0.1 (x-axis, gray arrow). In panels B, C, D changes in transmission scenarios relative to (**a**) are indicated with red arrows or removal of an arrow in the inset figure. Scenario 1 (**a**) explores a symmetric transmission rate for all species ($\beta_{SS} = \beta_{SC} = \beta_{CC} = 2.6 \times 10^{-1}$; $\beta_{CS}$ varies 0–0.26). Scenario 2 (**b**) explores a symmetric transmission rate within each species ($\beta_{SS} = \beta_{SC} = 2.6 \times 10^{-1}$; $\beta_{CC} = 3.0 \times 10^{-3}$; $\beta_{CS}$ varies 0–0.26).). Scenario 3 (**c**) explores asymmetric transmission rate between species ($\beta_{SS} = 2.6 \times 10^{-1}$; $\beta_{SC} = \beta_{CC} = 3.0 \times 10^{-3}$; $\beta_{CS}$ varies 0–0.26).). Lastly, scenario 4 (**d**) explores asymmetric transmission rate within and between species ($\beta_{SS} = 2.6 \times 10^{-1}$; $\beta_{SC} = 1.4 \times 10^{-1}$; $\beta_{CC} = 0$; $\beta_{CS}$ varies 0–0.26).). See Methods, Text S5, Text S6, Table S2 for more information on parameter values selected and model code. See Figs. S15–16 for results considering infectious periods of 8 and 14 days. Image Icon Credits: Cliparts Zone (sheep[76]), GetDrawing.com (goat[73], cattle[74]).

use in sheep and goats but to our knowledge has not yet been validated for use in cattle.

## Implications of cattle transmission to small ruminants for eradication

Transmission from host species that are not targeted for control will increase the multi-species reproductive number $R_0$ and impede eradication efforts. Therefore, we developed a general cross-species SIR modeling framework to explore the impact of not vaccinating cattle on the rate of small ruminant vaccination needed to bring the effective community reproductive number $R_E$ below 1 across varying cattle-to-small ruminant transmission scenarios. For PPRV, simulations from this framework suggest that an increasing level of small ruminant vaccination would be needed to reduce $R_E < 1$ as transmission from the non-target host cattle increased in two of the four transmission scenarios explored (Fig. 3b, d), rising from low levels to 40–60%

coverage needed. In one scenario (Fig. 3c), a small level of small ruminant vaccination may be sufficient across a wide range of increasing cattle transmission rates. In most realistic transmission scenarios (Fig. 3b–d), eradication is possible. However, if PPRV were to gain transmission competency in cattle and then transmit to small ruminants and other cattle at the same rates that small ruminants transmit PPRV (Fig. 3a), no amount of small ruminant vaccination is predicted to bring community $R_E < 1$.

Although these trials were designed with a large sample size to be able to detect low cattle-to-goat transmission, a limitation of this study is the use of a single PPRV lineage IV isolate. While PPRV/Ethiopia/Habru/2014 is from a lineage commonly circulating in Ethiopia[53], where PPR is endemic, and more broadly in Africa[54], PPRV lineages may differ in their ability to shed infectious virus from atypical hosts and start transmission chains. Additionally, we delivered a single dose and the size of dose was not varied; in the field, both may vary. However, PPRV/Ethiopia/Habru/2014

demonstrated key characteristics observed in the field including development of conventional clinical signs of appropriate severity in small ruminants, the ability to transmit between goats and cattle, and cattle seroconversion. As no cattle-to-goat transmission was observed, cattle-to-cattle transmission was not explored on the basis that sentinel goats would have provided the greatest chance of detecting PPRV transmission from cattle. Additionally, host species origin of the strain, host breed, and additional stressors (e.g., environmental stress, coinfection) may impact infection and transmission, and could be explored in future long term field trials. Finally, as measurements of inter- and intra-species transmission rates and host mixing patterns improve, the SIR modeling framework presented will need to be updated.

## Concluding remarks

The critical insight to arise from this study toward the goal of 2030 PPR eradication is that the most common cattle species found in sub-Saharan Africa (Zebu) are unlikely to be involved in transmission of PPRV to small ruminants. This study shows that, under co-housing conditions representing mixed species zero-grazing husbandry practices found in East Africa, cattle can become infected but do not transmit PPRV to goats. Given the experimental sample size, this study had an 80% chance of detecting a cattle-to-goat transmission event with an event probability as low as 0.05. No transmission was observed. Additionally, statistical analysis estimated a cattle-to-goat transmission probability of $2.94 \times 10^{-2}$ (95% CrI: $7.67 \times 10^{-4}$, $1.06 \times 10^{-1}$), which was nearly 32 times smaller than the estimated small ruminant-to-small ruminant transmission probability. This demonstrates that cattle-to-goat transmission, if it exists, is rare, including when animals are co-housed for an extended period. This finding supports the alternative hypothesis that cattle act as dead-end hosts in the epidemiology of PPR and currently have a negligible role as reservoirs of PPRV transmission to sheep and goats. Additionally, by using a SIR modeling framework with biologically realistic parameters, this study found that bringing the community $R_E < 1$ should be achievable with only sheep and goat vaccination in most realistic scenarios. However, if PPRV were to gain the ability to transmit from cattle hosts, intervention strategies would need to be revisited. For now, continued monitoring of circulation among potential non-target host species is warranted.

Going forward, a key test of the hypothesis that small ruminants are the only maintenance hosts for PPRV will be large-scale monitoring of PPRV circulation in the community of host species carried out alongside mass vaccination campaigns. During the early phase of the rinderpest (RP) eradication campaign, similar questions around wildlife reservoirs were posed and definitive confirmation that cattle were the sole maintenance host population was obtained only through observation of RP virus (RPV) elimination in wildlife following mass vaccination of cattle[55]. For PPR, a further important consideration for monitoring morbillivirus circulation at a population scale is that successful PPR eradication and the loss of host population immunity to PPRV may result in the vacant ecological niche[56] being occupied with another morbillivirus[57–60] through immunological competitive release. A similar explanation has been identified for monkeypox in filling the immunological niche created by the cessation of smallpox vaccination[61]. This process may have already contributed to successful PPR spread after RP eradication, and may already be underway with primary canine distemper virus (CDV) infection among cattle in northern Tanzania[60]. As PPR eradication efforts continue[10], selective pressure could result in novel PPRV mutations that enable more efficient replication and transmission from currently unrecognized livestock or wildlife hosts to be favored, as was predicted[62] and seen in suspected spillover of SARS-CoV-2 into wildlife hosts such as white-tailed deer[63] and mink[64] or highly pathogenic avian influenza into mink[65]. Broadly, morbilliviruses are adaptable with only a few amino acid changes enabling host range expansion in nature and in-vitro for CDV[66–68], and only one amino acid change enabling PPRV to enter human SLAM receptors and evade anti-Measles antibodies in-vitro[50]. Additionally, selective pressure could result in either increased virulence or decreased virulence in main hosts, the latter

of which was seen in the reduced virulence of RPV lineage 2 in cattle in the final stages of RP eradication[69–71]. Therefore, PPR molecular epidemiology studies should include whole genome sequencing and clinical sign documentation of circulating PPRV strains to detect and monitor mutations that could potentially increase viral fitness (e.g., increased replication and shedding from cattle) or adaption to other atypical hosts. Such work should be combined with animal movement data[72]. Importantly, continued vigilance is required and monitoring PPRV circulation and evolution across hosts[9] is a critical area for investment during PPR eradication.

## Inclusion & ethics statement

PPR is endemic in Ethiopia and has a major impact on small ruminant production and human livelihoods and investigating other potential hosts for PPRV that need to be considered for control was considered a relevant topic for study by AHI leadership. AHI researchers and other relevant local Ethiopian and international PPR researchers were included, consulted, and cited in all stages of the research process including study design, implementation, sample collection, processing, testing, analysis, manuscript editing, and as co-authors on this publication. Those who made valuable contributions to the project but did not meet the criteria for authorship are named in the Acknowledgements. All researchers were informed of personal risks and appropriate personal protective equipment and training for use and biosafety and security practices were provided and adhered to. Local researchers are in possession of all samples collected and a copy of all data produced and have ownership of both. As part of this study, infrastructural, laboratory, and analytical capacities were supported and expanded through the following: construction improvements to the biosecure animal facility, introduction and training on protocols for managing and investigation with new cell lines (VDS), molecular equipment maintenance, established collaborations with researchers in US and UK, and funding for local researchers to attend bioinformatics and epidemiological modeling trainings. Additionally, prior to the study, funded training was provided by the Association for Assessment and Accreditation of Laboratory Animal Care (AAALAC) in preparation for future certification. This research study was held to international animal welfare and biosafety standards and approved by the Animal Health Institute's Animal Research Scientific and Ethics Review Committee (ARSERC) under reference ARSERC/EC/001/17/04/2019.

## Data availability

Whole genome sequence information for PPRV/Ethiopia/Habru/2014 passage 4 can be found at Genbank accession number: ON110960. Clinical, serological, molecular, and tissue culture data from main and Supplementary Figs. are available in the following repository: https://github.com/cherz4/pprv-cattle-transmission, https://doi.org/10.5281/zenodo.12791123.

## Code availability

Code is available on GitHub: https://github.com/cherz4/pprv-cattle-transmission with doi for release v1.0.0: https://doi.org/10.5281/zenodo.12791123.

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

## Acknowledgements

The authors acknowledge the Supporting Evidence Based Interventions (SEBI) Project based at the University of Edinburgh and project management assistance from Mike Christian. We thank John Anderson for helpful study design and methodology conversations. We deeply thank our dedicated animal attendants Yohannis Gossa, Kassahun Mekonnen, Kelifa Ismaiel, Abebe Geresu, Gemechu Gebre, and Ejigu Zeben (supervisor) for their diligence in animal husbandry and sample collection. We also acknowledge the assistance of AHI IT Director Tewodros Alemu Gezaw in data management and the AHI animal procurement teams and finance office for their assistance on various project administration tasks. We thank the AHI sample reception staff for their assistance in organizing samples and entering them into the laboratory information management system. We thank Dr. Clint Leach for his advice and assistance in incorporating a Bayesian modeling approach to link empirical results to the math model. We thank Dr. Lindsay Beck-Johnson for helpful discussion of the model. We thank our reviewers for thoughtful comments and astute observations that greatly improved the manuscript. C.M.H., V.K., P.H., I.C., O.B. were supported by the Bill & Melinda Gates Foundation grant 'Program For Enhancing the Health and Productivity of Livestock (PEHPL)' (OPP1083453). C.M.H., F.A., D.Si., R.B., A.A., M.K., D.Sh., H.A., J.B., M.S.F., S.G., T.R.C., M.B., V.K. were supported by supplemental funding from the Bill & Melinda Gates Foundation through Supporting Evidence Based Interventions (SEBI) Project ROSLIN 2340. A.P. was supported by Supporting Evidence Based Interventions (SEBI) Project. V.K. was supported by the Penn State Huck Institutes of the Life Sciences Distinguished Chair. B.J.W. and S.C. were supported by the Biotechnology and Biological Sciences Research Council (Grant BB/R004250/1). The funders had no role in the study design, data collection and analysis, decision to publish, or the preparation of the manuscript.

## Author contributions

Conceptualization: C.M.H., N.J., V.K., S.C.; data curation: C.M.H., F.A.; formal analysis: C.M.H., O.N.B., V.K.; funding acquisition: A.P., C.M.H., J.B., I.M.C., P.J.H., V.K.; investigation: A.A.M., C.M.H., D.Sh., D.Si., H.A., F.A., M.K., R.B.; methodology: B.W., C.M.H., C.S., D.B., D.Si., F.A., M.B., O.N.B., V.K.; project administration: A.P., C.M.H., D.Si., F.A., J.B., M.S.F., T.R.C., S.G., V.K.; resources: C.M.H., D.Sh., D.Si., F.A., M.B., R.B.; software: C.M.H., O.N.B.; supervision: C.M.H., M.B., M.S.F., T.R.C., S.G., V.K.; validation: C.M.H., V.K.; visualization: C.M.H.; writing— original draft: C.M.H.; writing—review and editing: all authors.

## Competing interests

The authors declare no competing interests.
