## [Peer Review File · Communications Biology]

Reviewers' comments:

Reviewer #1 (Remarks to the Author):

Manuscript No.: COMMSBIO-23-1605-T

Comments to the authors:

The manuscript entitled "Empirical and model-based evidence for a negligible role of cattle in peste des petits ruminants transmission and eradication" is well written and the topic is novel and of high interest to researcher of PPR. The manuscript address a study with great importance to the current eradication program of PPR.

The manuscripts can be accepted for publications in Communications Biology after the author's make amendments that are necessary for improvement of the manuscript.

General comments: please be careful when using the term for the virus (PPRV, RPV) OR the disease (PPR, RP).

Line 7: Andrew R.Peters (affiliation no 7 not 8)

Line 7-8: Joram Buza (affiliation no 8 not 7)

Keywords:

Line 68: small ruminants morbillivirus is the species name but Not the virus name. please replace "small ruminants morbillivirus" by "peste des petits ruminants"

Abstract:

Lines 80-81: Did you use only goats and cattle for the experiment? Did you also use sheep? Because sheep is mentioned in the methods section but not in other sections.

Significance statement:

Line 100: please correct "show" to "showed".

Line 101: please correct "current eradication efforts" to "the current eradication program".

Main text:

Introduction:

Line 125: the correct classification is the family + genus + species Not only the species. please correct to "(PPRV; family: Paramyxoviridae; genus: Morbillivirus; species: *Small ruminants morbillivirus*). And please write all scientific names in Italic.

Line 127: please correct to "the World Organization ...".

Line 139: the eradication is for the disease Not the virus. please correct to "... insufficient to eradicate PPR .."

Line 141: the epidemiology is for the disease not the virus. please correct to "... and wildlife species in PPR epidemiology ...".

Results and Discussion:

Line 184: please correct to "The PPRV/Ethiopia/Habru/2014 isolate".

Line 185: please correct to "... with 4 inoculated goats were seroconverted, ".

Line 187: please correct to "..., so cattle were exposed to the excreted virus ...".

Line 192: "... were negative when virus isolation in cell culture was attempted".

Lines 190-195: did the ocular swabs from calves were tested by ELISA?

Line 204: delete the dot before the reference number in brackets.

Line 221: please correct to "None of the inoculated cattle shed PPRV ...".

Line 222: please correct to "Nasal and rectal swabs from the inoculated cattle were positive ...".

Line 228: please add "isolate" after the virus name.

Lines 229-230: please correct to "... experimental animals became infected as well as all 6 in-contact ...".

Concluding comments:

Line 303: please correct to "... hosts in the epidemiology of PPR ..".

Lines 313-314: please correct to "... the rinderpest (RP) eradication campaign, ...". RP Not RPV.

Line 316: please correct to "... through observation of RP elimination ...". RP Not RPV.

Lines 317-318: please correct to "For PPR, , a further important consideration for monitoring morbillivirus circulation at a population scale is that successful PPR ...". PPR Not PPRV.

Line 322: please correct to "... to successful PPR spread after RP eradication, ...".

Line 324: please correct to "As PPR eradication efforts ...".

Line 330: please correct to "in the final stages of RP eradication".

Line 336: please correct to "... investment during PPR eradication".

Materials and Methods:

Line 345: CHS-20 cells stands for what?

Line 345: please correct to "... cells expressing goat SLAM receptor".

Lines 347-348: please correct to "... grown on Vero-dogSLAM ..".

Line 368: please delete the dot before the reference number in brackets.

Line 374: please correct to "... post-infection ...".

Line 375: please correct to "... were allowed for free movement ...".

Line 378: please correct to "Samples (whole blood, serum, and swabs; ocular, nasal and rectal."

Line 391: please correct to "... reverse-transcription ...".

Line 393: please correct to "... tested in duplicates ...".

Line 396: please indicate the company from which the sandwich ELISA kit was purchased? Why the authors did not use the IC-ELISA from IDvet?

Lines 402-404: why the authors did not continue to collect the swabs from the group control goats?

Lines 408-409: please correct to "... or antigen sandwich ELISA were tested for infectious virus by inoculating into Vero-dogSLAM (VDS) cells".

Reviewer #2 (Remarks to the Author):

In this manuscript, the authors present an experimental infection study aiming to assess whether cattle can become infected with Peste des Petits Ruminants virus (PPRV), and transmit the virus to small ruminants. The lack of contribution of cattle to the endemic circulation of PPRV is generally accepted, and the results provide further evidence for this. A mathematical model is used to simulate PPRV transmission within and between species and assess the proportion of the small ruminant population that would need to be vaccinated to eliminate the disease under different assumptions about the infectiousness of cattle. I have several major comments:

One of the main results featured in the paper is that the estimation of “an upper bound for cattle-to-goat spillback transmission of 5%”. This percentage is not clearly defined in the manuscript and is likely to be misinterpreted by readers. In the appendix, it is suggested that under the experimental design, there was a ~80% chance to observe at least one cattle-to-small ruminant transmission event if the transmission probability was equal to 5%. This is not the same as an upper bound of the transmission probability. It would have been more useful to estimate the actual distribution of the probability of transmission given the experimental results.

The modelling of PPRV transmission could have been an important aspect of the manuscript, differentiating it from former papers that already provided evidence for the limited role played by cattle in PPRV transmission. However, the results of the experiment are not used to parameterise the model, and the analysis itself is very generic. While it is mentioned that the community R_0 value is 3.17 in the absence of control, this is not the case for most (if not all) scenarios investigated here. Additionally, no justifications are provided for the parameter values. The birth and mortality rates seem unrealistically high, and the uncertainty in R_0 would have justified the exploration of different scenarios.

The argument that “the models highlight the risk to PPR eradication strategies if virus strains with transmission competency in cattle were to emerge and emphasize the importance of continued monitoring of PPRV circulation and evolution across all known host species” is not convincing. There was no need for a model to “highlight” this, as the authors do not provide any clear argument related to the actual biology of the virus or closely related virus to support this claim. Giving it so much importance as one of the main insights of this work seems unjustified. The conclusion that “there is no need to modify the PPRV eradication campaign to include control measures for cattle” is more supported by the work presented in this paper and should be further emphasised, especially in the abstract.

Some additional comments:

Lines 165-166: “The trials were designed to emulate mixed animal husbandry practices observed in East Africa”: what does this involve exactly, beyond putting goats in contact with cattle in the same barns?

Line 185: you present results before explaining the design of the trial, even briefly, making it difficult for the reader to interpret them.

Lines 201-206: you report studies that estimated very different seroprevalence levels in cattle. Possible reasons for such different patterns need to be further discussed. For instance, were these studies conducted in different regions/farming systems where exposure of cattle to small ruminants or PPRV prevalence/seroprevalence in small ruminants are known to differ? Are these associated with different viral lineages/strains, and if yes, was the strain used in this experiment associated with high or low seroprevalence in cattle in the field?

Line 228: the authors describes another experiment, but it has not been clearly presented, making the reading confusing

Line 120: In “(...) combining empirical and computational approaches (...)”, do you mean “experimental” instead of “empirical”?

Line 142: “domestic livestock” delete “domestic”

Line 144: "domestic cattle" delete "domestic"

Reviewer #3 (Remarks to the Author):

Paper review: Empirical and model-based evidence for a negligible role of cattle in peste des petits ruminants transmission and eradication

The studies described in this paper aim to address a very important knowledge gap regarding the susceptibility of cattle to PPRV infection and clinical disease, and their role in virus transmission. There have been very few previous experimental studies that examine this, and this paper makes a very important contribution to the evidence that cattle are likely a dead-end host for PPRV. This is important for the PPR Global Eradication programme in which the main control measure is vaccination of sheep and goats only.

However, I found the paper difficult to navigate. The methods and results are not presented in a clear and logical order either in the main text or in the supplementary information – it was quite difficult to find out what was done in each trial, and the results of each trial. Some results and references to results in supplementary information are mixed up with the methods. I suggest that for clarity, in the methods section you summarise each numbered trial separately, indicating the aim of the trial, the numbers of animals of each species inoculated, in-contact and controls, and the number of replications. Further detail of methods can then be provided in the supplementary information. Similarly, the results section would be easier to navigate if the results of each numbered trial are presented in logical order, referencing further details in the supplementary information. Similarly, for figures indicate the trial number in the figure titles. Using the trial

numbers throughout will improve readability, which is important because this paper will be of interest to people from different disciplines and from within and outside academia.

The discussion of the results could be strengthened to better understand to what extent these results are generalisable to mixed cattle-small ruminant production systems in PPR endemic areas in Africa and Asia.

There is no discussion of what is known about the biological differences between cattle and small ruminants that make cattle less susceptible to infection, clinical disease and capacity to transmit? Was ethical approval obtained for this study?

Some more specific comments are provided below, using line numbers from the pdf of the manuscript.

Title: insert “virus” after “peste des petits ruminants”

Abstract:

Line 77: rather than saying that the role of cattle is unknown, which suggests no evidence, it is probably more correct to say that there is limited evidence that suggests that cattle do not play a significant role in PPRV circulation. This paper makes an important contribution that supports the findings of previous studies.

Line 79: “mimicked East African husbandry practices” Livestock production practices vary widely across East Africa and the husbandry practiced in these trials is not common, and would only occur in sedentary zero-grazing systems. However, the study housing provides a situation of intense contact between species that is much greater than is encountered in most East African husbandry systems, therefore providing a worst-case scenario – if transmission does not occur in this situation it is less likely to occur under field conditions. Perhaps you could say that the housing conditions provided inter-species contact that was similar to or greater than that occurring in the different types of East African husbandry systems.

Lines 80-83: consider rephrasing for clarity of which species showed clinical signs and which species sero-converted – at the moment the first sentence implies that neither species showed clinical signs or sero-converted. In the second sentence I think you are talking about the same 32 inoculated cattle? Give the number of cattle exposed to how many inoculated and exposed goats.

Line 87: experimental?

Line 87: negligible role – suggests the probability is so low that it probably never occurs. However, as you state later in the paper, the study only used one PPRV virus isolate and one type of cattle, but different PPRV isolates have different levels of pathogenicity in sheep and goats, and different breeds of sheep and goats have different levels of susceptibility. It is possible that there is similar variation for cattle, so a more nuanced conclusion would be better.

Significance Statement

Line 98-99: “..mixed species husbandry settings ..”

Introduction

Line 166: see previous comment re line 79.

Results and Discussion

Lines 183-184: “cattle ... do not develop clinical disease.” You state later (lines 188-190, line 198) in this paragraph that one calf had mild transient clinical signs, so the initial statement is not correct. In addition, this is based on one virus isolate and one cattle breed, so the statement needs to be careful not to overstate the results.

Lines 198-210: the discussion could include some more explanation and supporting references of the how intensity of contact between cattle and small ruminants varies in different mixed species husbandry systems, which could account for the variation in exposure to PPRV. As mentioned previously, the experimental conditions provided much greater contact than would be experienced in most husbandry systems.

Lines 209-210: what is meant by “extended shedding of PPRV from infected sheep and goats”? What environments favour extended PPRV survival?

Line 217-218: This statement seems to be incorrect: as before, some cattle showed transient clinical signs (line 220-221), and Figure 2 shows that one of the goats sero-converted, and one or more goats showed some pyrexia, and clinical signs up to score 6 at around day 12 - although combining the cattle and goat temperature and clinical scores in the same graphs with similar colours makes it difficult to tell.

Line 228-230: this could be stated more clearly by indicating the trial numbers, and specifying species. “in-contact positive control goats” – better to use the same terms consistently throughout the paper for each category of experimental animal – I think these are called sentinels elsewhere.

Line 230-234: see above re lines 217-218.

Line 239: for context, how does this likelihood compare to the likelihood of goat-goat transmission?

Line 253-255: can you explain more clearly why you conclude that this is “highly likely”.

Lines 255-257: what are the implications of this for your results? More false positives or more false negatives?

Line 263: what is meant by “non-equal vaccination rates”?

Line 265: is targeted vaccination modelled, or is it mass vaccination?

Line 269: what were the highest rates of spillback/ R_0 used in the model, to compare to Line 267?

Line 277-280: this is a bit unclear – lineage IV is common across Asia and Africa, but lineages I-III occur in different parts of Africa. More recent references required. However, is pathogenicity associated with lineage, or does it vary within lineage? Reference evidence from studies in small ruminants.

Lines 260-273: discuss the limitations of the model.

Line 281: would you consider the dose used to be high or low? How did it compare to the doses used in previous cattle trials?

Line 284: see above, there seems to be evidence of cattle-goat transmission.

Line 289: what is meant by “as measurements of inter- and intra-species transmission rates and host mixing patterns improve”? Do you mean if future experimental studies are carried out?

Figures 1-3. The figure titles would be more informative if they described what is being presented in the figure, rather than an interpretation of the results, and included the trial number.

Consider a greater colour contrast between groups or separate graphs for each group e.g. in figure 1 it is difficult to differentiate between the individual animal lines, especially the inoculated versus in-contact goats. The x axis is very compressed – consider extending this, or providing a larger version in the supplementary information. Variability among individuals within the groups is of interest but can't be distinguished.

Figure 3. mentions parameter values – but these are not specified anywhere in the paper.

Concluding comments

Line 297: zebu is a species of cattle not a cattle breed – there are many different breeds of zebu cattle in Africa.

Line 313-316: more relevant to this discussion is that rinderpest virus also infected sheep and goats, but rinderpest was eradicated without any intervention in small ruminants.

Line 324-328: do morbillivirus mutate at a similar rate as coronaviruses and influenza viruses?

Line 333: what is meant by “PPRV exit”

Materials and Methods

Lines 360-364: this seems to be results rather than methods.

Line 372: how was the cattle dose determined?

Line 377: did you monitor body temperature for a period of time prior to the trials to obtain a baseline?

Line 445: there is no Figure 4.

Line 445-448: describe the four transmission patterns that were explored. Provide a table of model parameters.

Supplementary Information

Figure S2. For clarity label trials 3, 4 and 5 individually.

Response to Reviewers for COMMSBIO-23-1605-T

We thank the reviewers for their time, comments, and attention to detail in providing this review. We address the comments below. Please note that due to other structural changes (e.g. moving methods before results) the original line numbers for these comments are now different starting in these sections and we have noted the new line numbers for new text added.

Reviewer 1.

The manuscript entitled "Empirical and model-based evidence for a negligible role of cattle in peste des petits ruminants transmission and eradication" is well written and the topic is novel and of high interest to researcher of PPR. The manuscript address a study with great importance to the current eradication program of PPR.

The manuscripts can be accepted for publications in Communications Biology after the author's make amendments that are necessary for improvement of the manuscript.

General comments: please be careful when using the term for the virus (PPRV, RPV) OR the disease (PPR, RP). Thank you for this comment, we have corrected all instances where this occurs.

Line 7: Andrew R.Peters (affiliation no 7 not 8) Corrected.

Line 7-8: Joram Buza (affiliation no 8 not 7) Corrected.

Keywords:

Line 68: small ruminants morbillivirus is the species name but Not the virus name. please replace "small ruminants morbillivirus" by "peste des petits ruminants" Corrected.

Abstract:

Lines 80-81: Did you use only goats and cattle for the experiment? Did you also use sheep? Because sheep is mentioned in the methods section but not in other sections.

Sheep were used in Trials 1-2 (see Text S1 Fig S2) to confirm that the PPRV field isolate caused clinical signs in sheep and goats and transmitted from goat-to-goat or sheep-to-sheep. The reviewer is correct, we tested cattle and goat transmissions to address the primary research question: starting with Trial 3, sheep were not included and only goat-to-goat, goat-to-cattle, and cattle-to-goat transmission were investigated.

To improve clarity, the following changes were made:

- Added "Sheep were not used after Trial 2" in the Animals and Study Design lines 221-222.
- Information on sheep has been removed from the main text with the exception of their brief mention in Trials 1-2 in the methods and to indicate that the isolate transmitted efficiently in the main hosts at lines 360-363.

Significance statement:

Line 100: please correct "show" to "showed". Corrected.

Line 101: please correct "current eradication efforts" to "the current eradication program". **Corrected.**

Main text:

Introduction:

Line 125: the correct classification is the family + genus + species Not only the species. please correct to "(PPRV; family: Paramyxoviridae; genus: Morbillivirus; species: Small ruminants morbillivirus). And please write all scientific names in Italic. **Corrected.**

Line 127: please correct to "the World Organization ...". **Corrected.**

Line 139: the eradication is for the disease Not the virus. please correct to "... insufficient to eradicate PPR .." **Corrected.**

Line 141: the epidemiology is for the disease not the virus. please correct to "... and wildlife species in PPR epidemiology ...". **Corrected.**

Results and Discussion:

Line 184: please correct to "The PPRV/Ethiopia/Habru/2014 isolate". **Corrected.**

Line 185: please correct to "... with 4 inoculated goats were seroconverted, **The suggested addition of 'were' does not fit our intended meaning (two cattle seroconverted). However we modified the rest of the sentence for improved clarity in lines 305-307.**

Line 187: please correct to "..., so cattle were exposed to the excreted virus ...". **Corrected.**

Line 192: "... were negative when virus isolation in cell culture was attempted". **Corrected.**

Lines 190-195: did the ocular swabs from calves were tested by ELISA? **Ocular swabs were not tested for any species by AgELISA. We have added this to lines 235-236.**

Line 204: delete the dot before the reference number in brackets. **Corrected.**

Line 221: please correct to "None of the inoculated cattle shed PPRV ...". **Corrected.**

Line 222: please correct to "Nasal and rectal swabs from the inoculated cattle were positive ...". **Corrected**

Line 228: please add "isolate" after the virus name. **Corrected, also added 'The' before the virus name.**

Lines 229-230: please correct to "... experimental animals became infected as well as all 6 in-contact ...". **Corrected**

Concluding comments:

Line 303: please correct to "... hosts in the epidemiology of PPR ..". **Corrected.**

Lines 313-314: please correct to "... the rinderpest (RP) eradication campaign, ...". RP Not RPV. **Corrected.**

Line 316: please correct to "... through observation of RP elimination ...". RP Not RPV. **Corrected.**

Lines 317-318: please correct to "For PPR, , a further important consideration for monitoring morbillivirus circulation at a population scale is that successful PPR ...". PPR Not PPRV. **Corrected.**

Line 322: please correct to "... to successful PPR spread after RP eradication, ...". **Corrected.**

Line 324: please correct to "As PPR eradication efforts ...". **Corrected.**

Line 330: please correct to "in the final stages of RP eradication". **Corrected.**

Line 336: please correct to "... investment during PPR eradication". **Corrected.**

Materials and Methods:

Line 345: CHS-20 cells stands for what? Adombi et al 2011 (ref 58) does not define the acronym (“This paper reports the development of a CV1 cell line stably expressing the goat SLAM protein. This new cell line, designated CHS-20, is highly efficient for isolating wild type PPRV from pathological specimens.”) We have changed lines 175-176 to “CHS-20 cells, which are (Flp-In-CV1) CV1 African Green Monkey kidney cells stably expressing the goat SLAM receptor”

Line 345: please correct to "... cells expressing goat SLAM receptor". Corrected, see above response.

Lines 347-348: please correct to "... grown on Vero-dogSLAM ..". Corrected.

Line 368: please delete the dot before the reference number in brackets. Corrected.

Line 374: please correct to "... post-infection ...". Corrected.

Line 375: please correct to "... were allowed for free movement ...". Modified to ‘were allowed to move freely’.

Line 378: please correct to "Samples (whole blood, serum, and swabs; ocular, nasal and rectal." Corrected.

Line 391: please correct to "... reverse-transcription ...". We do not believe the hyphen is common usage of this term so we have left the original text.

Line 393: please correct to "... tested in duplicates ...". We do not believe duplicate should be pluralized in this case so we have left the original text.

Line 396: please indicate the company from which the sandwich ELISA kit was purchased?

Why the authors did not use the IC-ELISA from IDvet? The authors did use the IDvet cELISA and IDvet IC-ELISA. All manufacturer names for all products used were listed in Text S2 – Supplemental Methods of the supplementary information. We have added more references to the methods supplement in more locations in main text.

Lines 402-404: why the authors did not continue to collect the swabs from the group control goats? After we confirmed that the biosafety and biosecurity protocols ensured that the negative controls in their own barn of the same building did not become infected or seroconvert (Trial 1), we stopped swab collection from negative controls in Trials 2-5 and we solely monitored clinical signs and seroconversion (on the same handling schedule as experimental animals) to save supplies. We clarified in lines 240-241.

Lines 408-409: please correct to "... or antigen sandwich ELISA were tested for infectious virus by inoculating into Vero-dogSLAM (VDS) cells". Corrected.

Reviewer 2:

I have several major comments:

One of the main results featured in the paper is that the estimation of “an upper bound for cattle-to-goat spillback transmission of 5%”. This percentage is not clearly defined in the manuscript and is likely to be misinterpreted by readers. In the appendix, it is suggested that under the experimental design, there was a ~80% chance to observe at least one cattle-to-small ruminant transmission event if the transmission probability was equal to 5%. This is not the same as an upper bound of the transmission probability. It would have been more useful to estimate the actual distribution of the probability of transmission given the experimental results.

We agree that our previously written statements were not the same as an upper bound on the cattle-to-goat transmission probability. We have adjusted this in all applicable areas throughout the manuscript (Abstract, Methods lines 193-195, Results & Discussion lines 363-365, Concluding Comments lines 433-438, Supplement). Furthermore, we have added code (Text S5) to estimate a confidence distribution (probability density function) around the probability of PPRV transmission (theta) based on the

experimental results ($n=32$, transmission = 0). We find that there is 0 probability that the probability of transmission is 0.27 or greater based on our results (new Figure S14). We have added text to the manuscript describing these in Methods lines 259-261, Results lines 365-368, and Conclusions lines 435-438.

The modelling of PPRV transmission could have been an important aspect of the manuscript, differentiating it from former papers that already provided evidence for the limited role played by cattle in PPRV transmission. However, the results of the experiment are not used to parameterise the model, and the analysis itself is very generic. While it is mentioned that the community R_0 value is 3.17 in the absence of control, this is not the case for most (if not all) scenarios investigated here. Additionally, no justifications are provided for the parameter values. The birth and mortality rates seem unrealistically high, and the uncertainty in R_0 would have justified the exploration of different scenarios.

We have made edits to improve clarity in this section.

- Clarifying text in the last paragraph of Methods, lines 284-297.
- Added a parameter table in the supplement (Table S2), indicating values across the four scenarios.
 - β s were set up in a matrix with values of 1, 2, or 3 based on scenario modeled (Fig 3 caption). In all scenarios small ruminant to small ruminant transmission $\beta_{SS}=3$ (the highest) and other values were adjusted to this to be equal to transmission among the main hosts or lower.
 - The μ rates were listed as 0.2 for small ruminants and 0.1 for cattle to indicate a higher background birth and mortality in small ruminants. Recovery rates γ were assumed the same between the species. The framework can be updated when there are improved estimates of these parameters available.
- Plausible values of the transmission rate β were generated by using the next generation framework, such that the combination of reasonable β , demographic (μ , birth and death rate), and recovery (γ) values resulted in a community R_0 value that fell between known published R_0 values that we previously cited in the text.
- Although we explore scenarios where there is control, we intentionally obtained reasonable parameter values in the absence of PPRV control as these reflect the PPRV transmission rate, as opposed to parameter values calculated from a control setting in which β may be biased downward and not reflect the transmission rate of PPRV.
- Uncertainty in R_0 is handled in the current framework.
 - In an SIR model of a completely susceptible population, $R_0 = \beta / \gamma$. Varying β as we did in each scenario varies R_0 (plotted as contours and shading), and explores uncertainty in transmission (and therefore uncertainty in R_0).
 - Due to our uncertainty in the correct transmission mode, we explored 4 scenarios that were the most plausible transmission mode scenarios.
 - Due to our uncertainty in inter- and intra-species transmission rates we took the preliminary step of obtaining reasonable β , μ , and γ values (which were used to calculate reasonable R_0 in observed ranges). We specifically highlight uncertainty in cattle to small ruminant transmission rates (β_{CS}) by exploring various values of β_{CS} in Fig 3 on the x-axis.

The argument that “the models highlight the risk to PPR eradication strategies if virus strains with transmission competency in cattle were to emerge and emphasize the importance of continued

monitoring of PPRV circulation and evolution across all known host species” is not convincing. There was no need for a model to “highlight” this, as the authors do not provide any clear argument related to the actual biology of the virus or closely related virus to support this claim. Giving it so much importance as one of the main insights of this work seems unjustified. The conclusion that “there is no need to modify the PPRV eradication campaign to include control measures for cattle” is more supported by the work presented in this paper and should be further emphasized, especially in the abstract.

We appreciate this point, and fully concur that the primary findings do not provide any evidence of a need to vaccinate cattle against PPR and this is emphasized in the results and abstract of the revised MS. We have also included text on biological mechanisms for pathogen evolution during vaccination campaigns, with a specific focus on morbilliviruses, into the Concluding Remarks (lines 464-467, 471-472). Specifically, we make clearer the molecular hurdles that PPRV and other morbilliviruses face for evolution and how few mutations it would take to enable entry into the cells of other host species and therefore expand host range.

We highlight the potential for a few number of changes in the primary structure of the virus to have big impacts on disease outcomes or host susceptibility. We note that PPRV may be restricted from exponential replication in the immune compartment (Tirumurugaan et al 2020), slowing down host disease and viral shedding. However a mutation could remove this restriction. It has been shown that a single amino acid substitution enables PPRV to enter human SLAMF1 receptors and evade anti-Measles virus antibodies (Abdullah et al 2018). Additionally, the replication machinery of PPRV works well in monkey cells in the lab (Bailey et al 2007) and in atypical host species (cattle – Mornet et al 1956, Asian water buffalo – Govindarajan et al 1997, camels – Saeed et al 2022, Mongolian saiga antelope – Pruvot et al 2020). Thus, PPRV is expected to be adaptable other host species with outcomes ranging from infection to clinical disease to mortality. More broadly, morbilliviruses have been shown to be highly adaptable (Benfield et al 2021, Ohishi et al 2014, Nikolin et al 2017, Bierginer et al 2013) and that, in the case of canine distemper virus (CDV) only 1-2 amino acid changes were needed for host range expansion.

Although the point is well taken that models may not be intuitively necessary here, our models show (and quantify) the consequences of any change that enables improved PPRV transmission from cattle. The models illustrate the thresholds at which if cattle were to gain transmission competency while maintaining ability to cause disease small ruminants we will not be able to eradicate PPRV by small ruminant vaccination alone. Given that cattle are under sampled in PPRV field studies and not tested for active infection, there is also uncertainty about which transmission mode scenario currently exists in the field. Hence, we propose continued monitoring of the potential for emergence of PPRV in cattle remains important.

Some additional comments:

Lines 165-166: “The trials were designed to emulate mixed animal husbandry practices observed in East Africa”: what does this involve exactly, beyond putting goats in contact with cattle in the same barns? This is correct. We were referring to the mixed species housing conditions often found in zero-grazing production systems in this region. We have modified all instances of husbandry practices to be more specific to production type and have also addressed this topic in response to Reviewer 3’s comments.

Line 185: you present results before explaining the design of the trial, even briefly, making it difficult for the reader to interpret them.

We have moved the Methods to occur before the Results.

Lines 201-206: you report studies that estimated very different seroprevalence levels in cattle. Possible reasons for such different patterns need to be further discussed. For instance, were these studies conducted in different regions/farming systems where exposure of cattle to small ruminants or PPRV prevalence/seroprevalence in small ruminants are known to differ? Are these associated with different viral lineages/strains, and if yes, was the strain used in this experiment associated with high or low seroprevalence in cattle in the field?

We agree that husbandry practices and production systems, livestock population turnover, and characteristics of circulating strains are important. We already noted several possible mechanisms, including management, for low and high seroprevalence patterns reported, in lines 324-328, 330-335, and 335-338. We clarified the geographic scope of the studies to be Asian and African countries (including: India, Sudan, Tanzania, Turkey, Ethiopia, Cameroon, and Mali) in lines in lines 322-323. Only one high seroprevalence study reported husbandry information (Pakistan, shared grazing and watering) whereas the other study speculated mixed free grazing in rangelands would yield the results observed among the purposively sampled clinically healthy cattle (Sudan).

Understanding the association between PPRV strains and high or low seroprevalence in the field requires further work. Most studies did not report association with a viral strain as they were solely serological studies, often of clinically healthy cattle, thus a range of different strains could have infected the sampled animals at different time points due to demographic or trade impacts on population turnover. It is not possible to link viral strains to lower or higher seroprevalence in the existing serological studies with the information available. No information for our study isolate PPRV/Ethiopia/Habru/2014 in cattle exists in the literature; however, we have added morbidity, mortality, and case fatality in sheep and goats for PPRV/Ethiopia/Habru/2014 (Alemu et al 2019) in lines 173-175.

Line 228: the authors describes another experiment, but it has not been clearly presented, making the reading confusing

Thank you for this comment. Our edits to the Methods mentioned in our response to Reviewer 1's comment on Lines 80-81 will help readers to find information on the additional experiments. Our response to Reviewer 1 explains why these experiments are not in the main text (as we believe this would be more confusing for the main focus of the paper, cattle-to-small ruminant transmission).

Line 120: In "(...) combining empirical and computational approaches (...)", do you mean "experimental" instead of "empirical"? For the title, and several instances in the introduction, we mean empirical, as in empirical evidence which complements evidence from modeling. We have checked each instance of empirical and experiment/experimental and either confirmed, changed, or removed the term throughout the manuscript.

Line 142: "domestic livestock" delete "domestic" Corrected.

Line 144: "domestic cattle" delete "domestic" Corrected.

Reviewer 3:

We appreciate the thoughtful and detailed feedback from this reviewer. We have addressed all comments with a response, but several are outside the scope for inclusion in the manuscript itself.

I found the paper difficult to navigate. The methods and results are not presented in a clear and logical order either in the main text or in the supplementary information – it was quite difficult to find out what was done in each trial, and the results of each trial. Some results and references to results in supplementary information are mixed up with the methods. I suggest that for clarity, in the methods section you summarise each numbered trial separately, indicating the aim of the trial, the numbers of animals of each species inoculated, in-contact and controls, and the number of replications. Further detail of methods can then be provided in the supplementary information. Similarly, the results section would be easier to navigate if the results of each numbered trial are presented in logical order, referencing further details in the supplementary information. Similarly, for figures indicate the trial number in the figure titles. Using the trial numbers throughout will improve readability, which is important because this paper will be of interest to people from different disciplines and from within and outside academia.

We agree, and we have moved the Methods before the Results and Discussion. We also understand that there was confusion between the order the trials were carried out and the manner in which they are presented / results are combined. Figure S2 associates trial number with experimental question, and we focus the main text on questions from Trials 3-5, across which we combined results of eight barns that each had 4 inoculated cattle to 4 sentinel goats. Thus, we left out trial numbers as they are not important for the conclusions in the main text regarding cattle and we anticipate will add to more confusion if included. (Please see response to Reviewer 1 for Lines 80-81).

The discussion of the results could be strengthened to better understand to what extent these results are generalisable to mixed cattle-small ruminant production systems in PPR endemic areas in Africa and Asia.

We agree and have addressed this in our response to Reviewer 3 Abstract Line 79 below. We have clarified throughout the manuscript that the husbandry in this study is similar to zero-grazing East African production systems.

For example, in Tanzania, calves are often kept together with small ruminants in the bomas until they are large enough to join the boma with the adult cattle. In Ethiopia, zero-grazing is practiced most often in highland areas where land resources are scarce due to crop dominant agriculture and dense human population. Pastoralists use boma practices (most often in lowland areas) and during drought the adult cattle and camels would be moved in search of feed and water, but sheep, goats and calves would be kept together locally in residential areas. So our husbandry practices are comparable to some practices in both sedentary zero-grazing and pastoralist production systems.

As Reviewer 3 points out, our study settings have the potential to have more intense contact - approaching conditions in market stalls or pastoralist bomas (where animals are kept safely at night). We suggest that our study settings represent an increasingly likely scenario, given the existing trend of rapid, increasing intensification in these production systems. Overall our study is much closer to field conditions than PPRV studies conducted in containment labs in the US or UK.

There is no discussion of what is known about the biological differences between cattle and small ruminants that make cattle less susceptible to infection, clinical disease and capacity to transmit?

These are areas of active research, and little is known at this time. We have cited the relevant literature in the main text.

Regarding cattle susceptibility to infection: To our knowledge, our study is the first to describe goat-to-cattle transmission (n=2) observed in experimental settings. We found that both in-contact cattle became infected. Often, cattle are not tested for PPRV, though the literature on PPRV serology in cattle has expanded in recent years and we have cited serological studies throughout the main text. We noted in the text various mechanisms for why cattle could be less susceptible than goats: cross-immunity from prior rinderpest vaccination (in older animals alive during vaccination campaigns line 452) or exposure to atypical morbilliviruses similar to rinderpest (Logan et al 2016), or canine distemper (Logan et al 2016) (line 326). Additionally there are no studies of PPRV binding kinetics to cells expressing cattle SLAM and nectin receptors, but a study (Fukuhara et al 2019) demonstrated that human Measles (MV) could use the canine SLAM receptor, but at a lower affinity, which may be important for productive infection.

Regarding cattle susceptibility to clinical disease: this has been noted in the literature, but not well studied (experimental studies: Mornet et al 1956, Schulz et al 2019, Couacy-Hyman et al 2019, and this study).

Was ethical approval obtained for this study?

Yes, this information was inadvertently left out in the submission and has now been added and addressed in the newly added Inclusion and Ethics Statement. Specifically, "This research study was held to international animal welfare and biosafety standards and approved by the Animal Health Institute's Animal Research Scientific and Ethics Review Committee (ARSERC) under reference ARSERC/EC/001/17/04/2019." (lines 499-502)

Some more specific comments are provided below, using line numbers from the pdf of the manuscript. Title: insert "virus" after "peste des petits ruminants" Corrected.

Abstract:

Line 77: rather than saying that the role of cattle is unknown, which suggests no evidence, it is probably more correct to say that there is limited evidence that suggests that cattle do not play a significant role in PPRV circulation. This paper makes an important contribution that supports the findings of previous studies. Corrected (ultimately removed due to abstract word limit).

Line 79: "mimicked East African husbandry practices" Livestock production practices vary widely across East Africa and the husbandry practiced in these trials is not common, and would only occur in sedentary zero-grazing systems. However, the study housing provides a situation of intense contact between species that is much greater than is encountered in most East African husbandry systems, therefore providing a worst-case scenario – if transmission does not occur in this situation it is less likely to occur under field conditions. Perhaps you could say that the housing conditions provided inter-

species contact that was similar to or greater than that occurring in the different types of East African husbandry systems.

We appreciate this point. We have clarified throughout the manuscript that the husbandry in this study is similar to zero-grazing East African production systems. (see response to General Comment above on same topic).

We agree our trial husbandry conditions could potentially allow for the intense contact scenario, given the zero grazing practice and ad libitum resource access. However, field examples such as the dense contact of a crowded market stall or a pastoralist homestead boma are likely to have more intense contact than the typical contact in our barns (see Fig S3 of animal facility). We followed international best practices to provide 3m³ of space per animal and our daily barn cleaning may not occur in the field and would impact environmental transmission. We assert that our study husbandry conditions represent an increasingly likely scenario, given the existing trend of rapid, increasing intensification in these production systems.

Lines 80-83: consider rephrasing for clarity of which species showed clinical signs and which species sero-converted – at the moment the first sentence implies that neither species showed clinical signs or sero-converted. In the second sentence I think you are talking about the same 32 inoculated cattle? Give the number of cattle exposed to how many inoculated and exposed goats.

Corrected (ultimately removed due to abstract word limit).

In the Results we have clarified that only the exposed cattle (n=2) showed mild transient clinical signs and seroconverted within 7-14 days whereas directly inoculated cattle remained asymptomatic and also seroconverted within 7-14 days.

Line 87: experimental? Corrected (ultimately removed due to abstract word limit).

Line 87: negligible role – suggests the probability is so low that it probably never occurs. However, as you state later in the paper, the study only used one PPRV virus isolate and one type of cattle, but different PPRV isolates have different levels of pathogenicity in sheep and goats, and different breeds of sheep and goats have different levels of susceptibility. It is possible that there is similar variation for cattle, so a more nuanced conclusion would be better.

We appreciate the reviewer's comment. Although we used a lineage IV strain believed to be widely representative (and highly transmissible and pathogenic), it is uncertain whether different PPRV strains in cattle may have different transmission outcome. One of the coauthors is currently investigating comparative pathology of different local Ethiopian PPRV isolates in local goats and cattle and results will be reported in a future publication.

Given other published experimental trials used other strains and breeds of cattle (Schulz et al 2019, Couacy-Hyman et al 2019), and they also found no transmission, we believe the use of 'currently play a negligible role' is appropriate. Although we cannot preclude that the combination of an alternative existing PPRV isolate or future PPRV isolate with the same or another cattle breed may show different results, we think our point is strengthened by an observation from the field. Evidence to date shows cattle seroconversion in the field but no evidence of increased PPRV transmission risk when cattle are introduced to a herd or other evidence of cattle-to-small ruminant transmission in the field, so we

believe cattle-to-small ruminant transmission is either currently not occurring or currently occurring at a very low level that is not detectable. Thus, we think the term 'negligible' is appropriate.

Significance Statement

Line 98-99: "...mixed species husbandry settings ..." Corrected.

Introduction

Line 166: see previous comment re line 79. Corrected, see responses to lines 79 and 98-99.

Results and Discussion

Lines 183-184: "cattle ... do not develop clinical disease." You state later (lines 188-190, line198) in this paragraph that one calf had mild transient clinical signs, so the initial statement is not correct. In addition, this is based on one virus isolate and one cattle breed, so the statement needs to be careful not to overstate the results. We have rephrased this to "*and can show mild, transient clinical signs.*" (lines 304-305). However, there was no transmission observed between cattle and goats, no live transmissible virus detected by isolation so we are careful not to suggest that the presence of these mild, transient clinical signs are due to transmission of virus or disease.

Lines 198-210: the discussion could include some more explanation and supporting references of the how intensity of contact between cattle and small ruminants varies in different mixed species husbandry systems, which could account for the variation in exposure to PPRV. As mentioned previously, the experimental conditions provided much greater contact than would be experienced in most husbandry systems.

We thank the reviewer for this suggestion. We have searched and have not found literature that describes the intensity of within-herd contacts in mixed species herds across husbandry systems. We agree these differences could have an impact but think an in-depth treatment of this is outside the scope of this study. For how husbandry practices across agropastoral and pastoral systems vary and impact PPRV transmission in mixed species herds in Tanzania, see Herzog et al 2020. While this does address husbandry practices, not much information on the intensity of contact can be gathered although herd size, use of mixed grazing, and presence of other species nearby are assessed via household survey. We assert that our study husbandry conditions represent an increasingly likely scenario, given the existing trend of rapid, increasing intensification in these production systems (see our response to Reviewer 3 Abstract Line 79 comment).

Lines 209-210: what is meant by "extended shedding of PPRV from infected sheep and goats"? What environments favour extended PPRV survival?

By extending shedding we mean shedding of PPR virus from a host that continues to occur beyond the typical timeframe of PPR disease course in that host. The conditions that favor extended PPRV survival in the environment and extended PPRV shedding are beyond the scope of this study, but the latter might include host stress including malnutrition or other infections. We mention this briefly as issues that may impact infection and transmission for future exploration in lines 333-336.

Line 217-218: This statement seems to be incorrect: as before, some cattle showed transient clinical signs (line 220-221), and Figure 2 shows that one of the goats sero-converted, and one or more goats showed some pyrexia, and clinical signs up to score 6 at around day 12 -although combining the cattle and goat temperature and clinical scores in the same graphs with similar colours makes it difficult to tell. We have clarified in the Abstract (removed due to word limit) and in Results (lines 309-311) which calf showed transient clinical signs. As noted in the text, cattle experimentally infected (inoculated) did not

show any clinical signs. Only one of the two calves that were infected naturally by exposure to infected co-housed goats showed mild, transient clinical signs.

Our statement that the inoculated cattle do not transmit PPRV to in-contact goats is also correct. The cattle did not shed PPRV RNA as detected by qRT-PCR, no PPRV RNA was recovered from goats for ocular, nasal, or fecal swabs, no goats seroconverted permanently (1/32 goats seroconverted transiently), and most goats showed no clinical signs and all goats did not observe clinical signs temporally consistent with an expected course of clinical disease.

For the observation of transient seroconversion for a goat and clinical scores > 0 in the figure:

- We are unable to explain the transient seroconversion in one goat (ID 173). This goat had a clinical score of 0 for the entire observation period with the exception of 10 dpi on which it was scored 1 due to a temperature score of 1 (> 39.5C but < 40 C). This goat was not in the same barn as goats with higher clinical scores. Additionally, the samples for this goat were tested in duplicate and then tested a second time in duplicate with the same qualitative findings (a period of transient short seroconversion).
- The three goats with the highest total clinical scores (3, 5, 6) all experienced this score only on 12 dpi in the same barn. This was the only score > 0 for two goats (166 and 181). Goat 176 scored a 1 on 7,10,35 dpi but otherwise a 0. So there was no clear progression of clinical signs consistent with a course of PPR disease for these goats whereas there was a clear progression of disease in the positive control barn (Fig S12A).
- Most goat observations with a higher clinical score on a given dpi had score of 1-2 in temperature, likely due to transient temperature spikes in the barn or an individual running higher than baseline for most of the observed period.

Line 228-230: this could be stated more clearly by indicating the trial numbers, and specifying species. “in-contact positive control goats” – better to use the same terms consistently throughout the paper for each category of experimental animal – I think these are called sentinels elsewhere. Corrected, we have replaced ‘in-contact’ terminology with sentinel terminology throughout.

Line 230-234: see above re lines 217-218.

Our response above for lines 217-218 addresses this comment. We have added also modified to “no lasting antibodies” at this location to increase clarity as well as “and showed minimal clinical signs not consistent with a typical course of disease”

Line 239: for context, how does this likelihood compare to the likelihood of goat-goat transmission?

We have replaced the two instances of ‘likelihood’ with ‘probability’.

To our knowledge, there have been no studies quantifying the likelihood of goat-to-goat or cattle-to-small ruminant transmission of diseases. Hence, we conducted a power analysis (detailed in Figure S1 and Text S4, including code) by simulating 1,000 scenarios from a binomial distribution. These simulations spanned a plausible range for cattle-to-goat transmission probabilities (0.01-10%) given our sample size of 32. The analysis aimed to determine the threshold probability of cattle-to-goat transmission at which there would be an 80% chance of detecting such an event. We found this threshold to be at a 5% transmission probability. We established an 80% detection probability as the minimum acceptable level in our study.

In the goat-to-goat transmission trials, we observed a 100% transmission rate in all co-housed inoculated-sentinel goat pairings. While this rate might be lower in field conditions, we anticipate it remaining significantly high in a fully susceptible herd. We conducted 10 trials with inoculated goat – sentinel goat pairs. A supplementary power analysis (not included in this manuscript), indicates that with our sample size, we would have an 80% chance of detecting goat-to-goat transmission for a transmission probability as low as approximately 0.15.

Line 253-255: can you explain more clearly why you conclude that this is “highly likely”.

We believe this explanation is highly likely as, once the experiment in any given barn started, after a few dpi PPRV would be shed by inoculated animals all over the barn and it is likely that any animal in this barn sniffing their feed or the ground or other infected surface could pick up PPRV in their nasal cavity that would be detected on a nasal swab by 7 and 10 dpi as mentioned on lines 314-315. We have added clarifying statements about what is/is not visualized in the main text at lines 316-317 and in supplement Text S2 Serological and Molecular Analysis subsection last sentence.

Lines 255-257: what are the implications of this for your results? More false positives or more false negatives?

Since the study was not designed to test the performance characteristics of the ELISA the implications of AgELISA's performance, we are unable to conclusively comment on assay's performance limitations in cattle. Our studies provide evidence of potentially reduced specificity since in some cases where AgELISA indicated viral presence, yet qRT-PCR, a more sensitive test, did not confirm these findings. Additionally, Schulz et al. (2019) presented results that suggest an increased risk of false negatives, particularly in scenarios with lower viral loads that qRT-PCR can detect but AgELISA might miss. Therefore, while AgELISA provided rapid preliminary insights in our trial, because of potential concerns with lower specificity and sensitivity that would confound the analysis because of false positive and negative results, we concluded qRT-PCR results should be regarded as more reliable and definitive for assessing PPRV presence in cattle within our research as has been done in this study.

Line 263: what is meant by “non-equal vaccination rates”?

Vaccination rates that are not equal; in this case sheep and goats will receive vaccination a part of the current eradication strategy at some positive, non-zero rate and cattle will not receive vaccination (rate: 0). We now clarify the text at lines 396-398.

Line 265: is targeted vaccination modelled, or is it mass vaccination?

Targeted mass vaccination of sheep and goats is modelled. We had been using the word targeted to refer to which species were targeted for control (vaccination) as opposed to a subset of the small ruminant population. We removed once instance of the word targeted at line 399 to improve clarity.

Line 269: what were the highest rates of spillback/R0 used in the model, to compare to Line 267?

We have rephrased and added the higher rates of transmission (beta) to lines 400-403.

Line 277-280: this is a bit unclear – lineage IV is common across Asia and Africa, but lineages I-III occur in different parts of Africa. More recent references required. However, is pathogenicity associated with lineage, or does it vary within lineage? Reference evidence from studies in small ruminants.

Some evidence exists on how PPRV lineages vary in pathogenicity (Couacy-Hymann et al 2007 and 2009) but less is known about variance within and between strains regarding pathogenicity, shedding, and host range. We provided older references that we think provided relevant context for lineage IV extent

across Africa (ref 41, 2011; also supported by Libeau et al 2014 though we did not cite it), but also included newer references (ref 40, 2019) about Ethiopia specifically.

Lines 260-273: discuss the limitations of the model.

We addressed model limitations in the limitations paragraph (lines 423-425), model assumptions (lines 277-278), and explanation of parameter values (lines 283-284). To our knowledge there are no empirical measurements of within-herd transmission rates (β) for small ruminant-to-small ruminant (intra-species), small ruminant-to-cattle (inter-species), or cattle-to-small-ruminant (inter-species) transmission. When these become available, they could be used within our modeling framework and confidence in the conclusions would be expected to increase (lines 423-425).

Line 281: would you consider the dose used to be high or low? How did it compare to the doses used in previous cattle trials?

We appreciate the need for considering the challenge dose used in our study relative to previous cattle trials. However, the challenge dose is highly context-dependent, making direct comparisons between studies difficult. Our study's primary focus was on a strain capable of productive infection and transmission to other hosts or species. We believe the appropriate level of comparison between studies is based on transmission potential (e.g., did it transmit?).

Directly addressing the dosage question: Couacy-Hymann et al. (2019) administered a 1mL subcutaneous inoculation in cattle and goats with a concentration of 10^3 TCID₅₀/mL, but detailed information on isolation or passage was not provided. Schulz et al. (2019) used a 2 mL dose (1 mL in each nostril) for cattle with a concentration of $10^{4.5}$ TCID₅₀/mL, using a strain initially isolated on CHS-20 cells and passaged twice on VDS cells. In our study, the viral concentration ranged between $10^{5.3}$ and $10^{5.6}$ TCID₅₀/mL. We administered 1mL to small ruminants and 2mL to cattle (1 mL in each nostril). The volume for cattle matches Schulz et al. (2019) and is double that of Couacy-Hymann (2019). Our concentration, however, is significantly higher than both – approximately 10 times that of Schulz et al. and about 400 times that of Couacy-Hymann et al., and it likely exceeds what an animal would inhale naturally in the field. The rationale for the higher volume in cattle was due to their larger body size compared to small ruminants. This approach was intended to create a highly conducive scenario for inducing clinical signs in main hosts and potentially in atypical hosts (cattle). Although we did not observe robust evidence of disease in cattle, this could be an area for future research, possibly by increasing the dose further to assess any host restrictions. However, as our current study aimed to establish clinical disease in sheep and goats with efficient transmission, and then examine natural infection pathways between cattle, sheep, and goats, exploring higher doses in cattle was beyond our scope.

Line 284: see above, there seems to be evidence of cattle-goat transmission.

See our response for Reviewer 3 Results and Discussion Line 217-218.

Line 289: what is meant by “as measurements of inter- and intra-species transmission rates and host mixing patterns improve”? Do you mean if future experimental studies are carried out?

We have addressed future studies as an option in our response to Reviewer 3 for Lines 260-273 above and in the main text at lines 423-425.

Figures 1-3. The figure titles would be more informative if they described what is being presented in the figure, rather than an interpretation of the results, and included the trial number. Consider a greater colour contrast between groups or separate graphs for each group e.g. in figure 1 it is difficult to

differentiate between the individual animal lines, especially the inoculated versus in-contact goats. The x axis is very compressed – consider extending this, or providing a larger version in the supplementary information. Variability among individuals within the groups is of interest but can't be distinguished. We thank the reviewer for this suggestion. We think the interpretation is helpful for the reader at the title but we point out that the rest of the caption describes what is being presented in detail.

We intentionally included individual animal lines at a much lighter transparency in favor of the clear darker lines that show smooth local regression (LOESS) curves of all animals in the group (control, inoculated, sentinel) and corresponding 95% confidence bands (grey shading).

Figure 3. mentions parameter values – but these are not specified anywhere in the paper.

Transmission (beta) parameters were mentioned in Figure 3 caption, the code (and parameter values) for the model and each panel are provided in Text S6 (previously Text S5). We have added a parameter table to the supplement (Table S2) to clearly show all parameter definitions and values.

Concluding comments

Line 297: zebu is a species of cattle not a cattle breed – there are many different breeds of zebu cattle in Africa. Corrected to species.

Line 313-316: more relevant to this discussion is that rinderpest virus also infected sheep and goats, but rinderpest was eradicated without any intervention in small ruminants.

We indicated that RPV was only eliminated in wildlife following the mass vaccination of cattle as we are confident wildlife were not vaccinated and we sought to make the point that we may similarly determine if PPRV can be eliminated in cattle or other atypical host species by solely conducting small ruminant vaccination. However, during the RPV eradication, small ruminants could still be infected and benefit from PPRV antibodies as protection from RPV. During the current PPRV eradication, cattle and other former RPV hosts will not have the benefit of RPV antibody protection against PPRV and it is unclear if antibodies from CDV infection in cattle could provide protection (Logan et al 2016). It is unclear if this difference in protection of the non-target host species cattle will result in the same outcome as during the RPV eradication campaign – e.g. that no control is needed in the non-target host species.

However, RPV vaccination in sheep and goats was effective for RPV protection and has been documented to have been widely used for PPR protection in some regions (eg Ethiopia Kock PhD thesis 2008, cited in Roeder et al 2013), though this appears to be near the end of the official eradication program so we agree that RPV was eradicated without significant intervention in small ruminants.

Line 324-328: do morbillivirus mutate at a similar rate as coronaviruses and influenza viruses?

This is an interesting question but outside of the scope of this study to address. Briefly, some published rates from PPRV, influenza A in birds, and SARS-CoV-2 in humans are listed.

The mutation rate for PPRV has been estimated as the following:

- Mahapatra et al 2021: 7.224×10^{-4} , 95% HPD interval 4.088×10^{-4} – 1.065×10^{-3} substitutions/site/year (HPD = highest probability density)
- Clarke et al 2017: 9.22×10^{-4} , 95% HPD 6.206×10^{-4} – 1.26×10^{-3}
- Sahu et al 2017: 7.684×10^{-4} , 95% HPD 7.233×10^{-4} – 8.1327×10^{-4}
- Muniraju et al 2014: 9.09×10^{-4} , 95% HPD 2.13×10^{-4} – 1.64×10^{-3}
- Adombi et al 2015: 7.8×10^{-4} , 95% HPD 7.3×10^{-4} – 8.4×10^{-4}

- Collectively, published and unpublished work we are aware of on PPRV puts the range between 5×10^{-4} and 10×10^{-4} .

Comparing to literature on other pathogens, influenza virus in birds mutates approximately $30\text{-}50 \times 10^{-4}$ and SARS-CoV-2 mutates at approximately the same rate as PPRV, at 5×10^{-4} substitutions/site/year.

Line 333: what is meant by “PPRV exit”

While we had originally meant PPRV exit from cattle cells, we do not think the entry exit terminology for PPRV and receptors is clear or appropriate here. We have removed it here and throughout the manuscript and edited lines 472-473. Current evidence suggests cattle can be infected with PPRV and seroconvert but PPRV replication is dramatically impacted outside the immune cells in cattle reducing clinical disease and capacity to transmit (Tirumurugaan et al 2020).

Materials and Methods

Lines 360-364: this seems to be results rather than methods.

We have moved this text to a new Methods subsection ‘Clinical validation of Isolate’ and added lines 217-219 for clarification. The remaining text in this section points readers to the supplement for results from Trials 1 and 2 and a supplemental challenge, which establish that the isolate has the expected clinical signs, transmission behavior, and immunological protection characteristics in the main host species. These findings form the background work and methods ahead of the experiments that include cattle, which are the only trials that address the main question of the paper. Additionally we direct the reader to our response to Reviewer 3 Supplemental Information comment on Figure S2.

Line 372: how was the cattle dose determined?

We have addressed this in detail our response to Reviewer 3 Line 281 comment above. We relied on evidence from experimental infection and natural transmission from goats to cattle and did not find robust evidence of disease in cattle so we did not further pursue an investigation of dosing in cattle as it was outside the scope of the study (which was to establish clinical disease and transmission in goats and sheep and examine natural transmission from goats to cattle and cattle to goats).

Line 377: did you monitor body temperature for a period of time prior to the trials to obtain a baseline?
Yes, we monitored body temperature for several days prior to the pilot trial, starting 4 days before.

Line 445: there is no Figure 4. Corrected to Figure 3.

Line 445-448: describe the four transmission patterns that were explored. Provide a table of model parameters.

The transmission patterns were described in Figure 3 caption. This information has been added to the Methods and edited for further clarity (lines 292-298).

Supplementary Information

Figure S2. For clarity label trials 3, 4 and 5 individually.

Furthermore, trial numbers have also been intentionally removed from the majority of the text to reduce confusion. Instead, we direct the reviewers and readers to Figure S2 which aligns questions asked with trial numbers. Trials numbers refer to experiments carried out at the same calendar time, but within a trial more than one question may have been tested in different barns, as was the case with

Trial 3 which tested goat-to-cattle transmission in one barn as well as cattle-to-goat transmission in another barn concurrently. The main finding on cattle-to-goat transmission came from a combination of eight barns across Trials 3, 4, 5. We believe this figure represents the best way to visually condense the experiments run in specific barns at specific times and trials numbers in a way focuses on results that address the questions asked by specific barns and trials.

Reviewers' comments:

Reviewer #1 (Remarks to the Author):

Manuscript No.: COMMSBIO-23-1605-A

Comments to the authors:

The manuscript entitled "Empirical and model-based evidence for a negligible role of cattle in peste des petits ruminants transmission and eradication" is well written and the topic is novel and of high interest to researcher of PPR. The manuscript addresses a study with great importance to the current eradication program of PPR.

The authors have improved the manuscript but it still needs a few important amendments to be done before the manuscript is accepted for publication.

These amendments are as follows:

General comments: please be careful when using the term for the virus (PPRV, RPV) OR the disease (PPR, RP).

This was not corrected as suggested

Abstract

Lines 71: "To investigate the role of cattle in the epidemiology of PPRV" please replace PPRV with PPR.

Introduction

Line 152: "could impact PPRV eradication efforts." please replace PPRV with PPR.

Line 156: "where PPRV is endemic." please replace PPRV with PPR.

Line 166: "there is no need to modify the PPRV eradication campaign" please replace PPRV with PPR.

Results and discussion

Line 322: "or when a PPRV epidemic spreads through a co-housed" please replace PPRV with PPR.

Line 412: "where PPRV is endemic" please replace PPRV with PPR.

Concluding remarks

Line 430: "2030 PPRV eradication" please replace PPRV with PPR.

Line 470: "Therefore, PPRV molecular epidemiology studies" please replace PPRV with PPR.

Inclusion and ethical statement

Line 483: "PPRV is endemic in Ethiopia" please replace PPRV with PPR.

Line 486: "local Ethiopian and international PPRV researchers" please replace PPRV with PPR.

Reviewer #2 (Remarks to the Author):

The manuscript has been enhanced in terms of clarity; however, my primary concerns raised during the initial review remain unaddressed.

I observe that the previously mentioned "upper bound for cattle-to-goat spillback transmission of 5%" has been now been replaced by an assertion that "statistical analysis of the observed experimental results indicates there is no probability that the probability of cattle-to-goat transmission is 0.27 or above." Nonetheless, this representation seems imprecise, given that this probability, while extremely low, is not zero. I would recommend, once more, that the authors deduce the posterior distribution of

the transmission probability. This would not only correct the current misrepresentation but also allow for direct incorporation of this distribution into the mathematical model.

Regarding the modelling analysis, it does not incorporate the empirical findings from the experiment and remains broadly generic. For instance, the observation that if $R_0 > 1$ in a given population (cattle here) then vaccinating an in-contact population (small ruminants) cannot reduce the community R_e below 1 is a straightforward theoretical result, and this modelling exercise was not needed to make this point. Some modelling assumptions are questionable (e.g. PPR not causing any death in small ruminants). Despite the inclusion of a table listing parameter values—a helpful addition—the justification for some parameter values is lacking, the birth and mortality rates appear implausibly high. This oversight is significant, suggesting an unrealistically rapid reduction of the recovered and immune animal populations.

Responses to Reviewers: Revision Round 2

Author responses are in green. We thank both reviewers for their time, helpful feedback, and attention to detail. We believe the manuscript is improved by this feedback.

Reviewer 1 Comments to the authors:

The manuscript entitled "Empirical and model-based evidence for a negligible role of cattle in peste des petits ruminants transmission and eradication" is well written and the topic is novel and of high interest to researcher of PPR. The manuscript addresses a study with great importance to the current eradication program of PPR. The authors have improved the manuscript but it still needs a few important amendments to be done before the manuscript is accepted for publication. These amendments are as follows:

General comments: please be careful when using the term for the virus (PPRV, RPV) OR the disease (PPR, RP). This was not corrected as suggested.

All of our corrections match the line numbers below (or are within 1-2 lines)

Abstract

Lines 71: "To investigate the role of cattle in the epidemiology of PPRV" please replace PPRV with PPR. **Corrected.**

Introduction

Line 152: "could impact PPRV eradication efforts." please replace PPRV with PPR. **Corrected.**

Line 156: "where PPRV is endemic." please replace PPRV with PPR. **Corrected.**

Line 166: "there is no need to modify the PPRV eradication campaign" please replace PPRV with PPR. **Corrected.**

Results and discussion

Line 322: "or when a PPRV epidemic spreads through a co-housed" please replace PPRV with PPR. **Corrected.**

Line 412: "where PPRV is endemic" please replace PPRV with PPR. **Corrected.**

Concluding remarks

Line 430: "2030 PPRV eradication" please replace PPRV with PPR. **Corrected.**

Line 470: "Therefore, PPRV molecular epidemiology studies" please replace PPRV with PPR. **Corrected.**

Inclusion and ethical statement

Line 483: "PPRV is endemic in Ethiopia" please replace PPRV with PPR. **Corrected.**

Line 486: "local Ethiopian and international PPRV researchers" please replace PPRV with PPR. **Corrected.**

Reviewer #2 (Remarks to the Author):

The manuscript has been enhanced in terms of clarity; however, my primary concerns raised during the initial review remain unaddressed.

I observe that the previously mentioned "upper bound for cattle-to-goat spillback transmission of 5%" has been now replaced by an assertion that "statistical analysis of the observed experimental results indicates there is no probability that the probability of cattle-to-goat transmission is 0.27 or above." Nonetheless, this representation seems imprecise, given that this probability, while extremely low, is not zero. I would recommend, once more, that the authors deduce the posterior distribution of the transmission probability. This would not only correct the current misrepresentation but also allow for direct incorporation of this distribution into the mathematical model.

Thanks for this comment – we agree that the probability, by definition, is not truly zero. We have now calculated the posterior distribution of the transmission probabilities for cattle-to-goat, goat-to-cattle, and goat-to-goat transmission and corresponding 95% credible intervals. We have replaced the wording (see highlighted text in manuscript in Methods and Results Lines 262-277, 299-304, 372-377, 387-392, 423-426, 459-461) with the mean from the posterior of cattle-to-goat transmission and corresponding 95% credible interval.

We have removed the confidence distribution code (Text S5) and plot (Fig S14) and corresponding text in the Results and Discussion. We have added code to calculate the posterior distributions, transmission rates and confidence intervals in Text S5 and a figure of the prior and posterior distributions in Fig S14 and updated the Results and Discussion text (highlighted same aforementioned line numbers) to provide revised transmission probabilities and transmission rates (Methods, Table S2, Figure 3 caption).

Regarding the modelling analysis, it does not incorporate the empirical findings from the experiment and remains broadly generic. For instance, the observation that if $RO > 1$ in a given population (cattle here) then vaccinating an in-contact population (small ruminants) cannot reduce the community Re below 1 is a straightforward theoretical result, and this modelling exercise was not needed to make this point. Some modelling assumptions are questionable (e.g. PPR not causing any death in small ruminants). Despite the inclusion of a table listing parameter values—a helpful addition—the justification for some parameter values is lacking, the birth and mortality rates appear implausibly high. This oversight is significant, suggesting an unrealistically rapid reduction of the recovered and immune animal populations.

We have revised the model code (Text S6) and parameter table (Table S2) so that the mathematical model uses transmission rates calculated from the mean of the posterior distributions derived from our trials (see Text S5, Figure S14 and line numbers in our response to the comment immediately above). We have also revised the mortality rates and recovery rates in Table S2 to have more realistic values (corresponding to reasonable small ruminant and cattle lifespans) and literature citation. Finally, Figure 3 and its caption and corresponding Methods and Results and Discussion text have been updated (highlighted) to describe the revised results.

Reviewers' comments:

Reviewer #2 (Remarks to the Author):

The revisions made to the manuscript are commendable. However, upon reviewing the new model results, there appears to be a potential issue. The current findings suggest that reducing R_e below 1 only requires vaccinating an extremely small fraction of the small ruminant population, unless there is significant viral transmission occurring with cattle. This observation seems questionable and does not align with observed PPRV dynamics in the field. If such a minimal vaccination coverage was effective, one would expect the disease to have been eradicated already given current vaccination efforts. Therefore, further clarification of these model results may be necessary to ensure their alignment with real-world epidemiological patterns.

Responses to Reviewers: Revision Round 3

Author responses are in green. We thank both reviewers for their time, helpful feedback, and attention to detail. We believe the manuscript is improved by this feedback.

Reviewer #2 (Remarks to the Author):

The revisions made to the manuscript are commendable. However, upon reviewing the new model results, there appears to be a potential issue. The current findings suggest that reducing R_e below 1 only requires vaccinating an extremely small fraction of the small ruminant population, unless there is significant viral transmission occurring with cattle. This observation seems questionable and does not align with observed PPRV dynamics in the field. If such a minimal vaccination coverage was effective, one would expect the disease to have been eradicated already given current vaccination efforts. Therefore, further clarification of these model results may be necessary to ensure their alignment with real-world epidemiological patterns.

We greatly appreciate Reviewer #2's astute observation regarding the implausible vaccination coverage suggested by our model results. We fully agree that the low vaccination requirement initially reported does not align with real-world epidemiological patterns for PPRV.

Upon reviewing our model, we identified an error in converting transmission probabilities to rates. Specifically, the transmission rates were previously calculated based on the duration of the trial (35 days) rather than the infectious period (estimated to be 10 days, consistent with both our observations and the literature). Correcting this conversion results in a significant adjustment in the required vaccination coverage.

Our updated model now indicates that achieving effective control requires vaccinating a substantially higher proportion of the small ruminant population. Specifically, our revised results show that a vaccination coverage of approximately 70% or higher is necessary, especially as the cattle-to-small-ruminant transmission rate increases (Figure 3, scenarios B, C, and D). This revised finding aligns much more closely with field observations and provides a realistic estimate consistent with the current understanding of PPRV epidemiology and vaccine coverage.

We have updated the manuscript to reflect these corrected results and sincerely thank the reviewer for bringing this issue to our attention. We have included a special acknowledgment in the manuscript expressing our gratitude.

REVIEWERS' COMMENTS:

Reviewer #2 (Remarks to the Author):

I agree that the newly presented transmission rate between small ruminants seems more plausible